# Uncertainty Estimation on Graphs with Structure Informed Stochastic Partial Differential Equations

**Fred Xu**
Block Inc
University of California, Los Angeles
fredxu@squareup.com,

**Thomas Markovich**
Block Inc
tmarkovich@squareup.com

## Abstract

Graph Neural Networks (GNNs) have achieved impressive results across diverse network modeling tasks, but accurately estimating uncertainty on graphs remains difficult—especially under distributional shifts. Unlike traditional uncertainty estimation, graph-based uncertainty must account for randomness arising from both the graph's structure and its label distribution, which adds complexity. In this paper, making an analogy between the evolution of a stochastic partial differential equation (SPDE) driven by Matérn Gaussian Process and message passing using GNN layers, we present a principled way to design a novel message passing scheme that incorporates spatial-temporal noises motivated by the Gaussian Process approach to SPDE. Our method simultaneously captures uncertainty across space and time and allows explicit control over the covariance kernel's smoothness, thereby enhancing uncertainty estimates on graphs with both low and high label informativeness. Our extensive experiments on Out-of-Distribution (OOD) detection on graph datasets with varying label informativeness demonstrate the soundness and superiority of our model to existing approaches.

## 1 Introduction

Uncertainty Estimation is crucial to developing reliable machine learning systems, now deployed in safety-critical fields such as healthcare [15], medicine [2], and financial modeling [14]. This task usually handles uncertainty from the lack of data samples and the inherent randomness in the data generating process [13] and has traditionally been addressed for i.i.d data using energy based models [44], Bayesian learning framework [6], or stochastic process to inject random noise into model learning [22, 47]. Already a challenging problem, Uncertainty Estimation when applied to graph data, requires even more consideration on the dependency between nodes and even substructures on a graph. For graph data, Graph Neural Networks (GNNs) have been the standard model, leading state-of-art performance on many graph related learning tasks. By aggregating information from neighbors, GNNs are able to learn meaningful representations of nodes efficiently [21, 50]. Despite their superior performance, GNNs can generate overly confident predictions on tasks even when making the wrong predictions [45]. Recently many works have attempted to address uncertainty estimation for graph data in the similar manners as i.i.d data, using the Bayesian framework [45], energy based model [12, 55], or through injecting noise in model training using a stochastic process [26, 3]. Despite some remarkable progresses, these models assume that nodes that are close-by in the graph should share similar uncertainty, which often connects to the concept of graph homophily[55, 26]. However, in cases such as low *label informativeness* [35], nodes that are close to each other often have different labels distribution patterns, resulting in the phenomenon that knowing the neighborhood labels doesn't decrease uncertainty of center node's label. [31, 32]. When graphs exhibit low label informativeness, tasks like node classification become more difficult for traditional GNNs, and so is uncertainty

39th Conference on Neural Information Processing Systems (NeurIPS 2025).

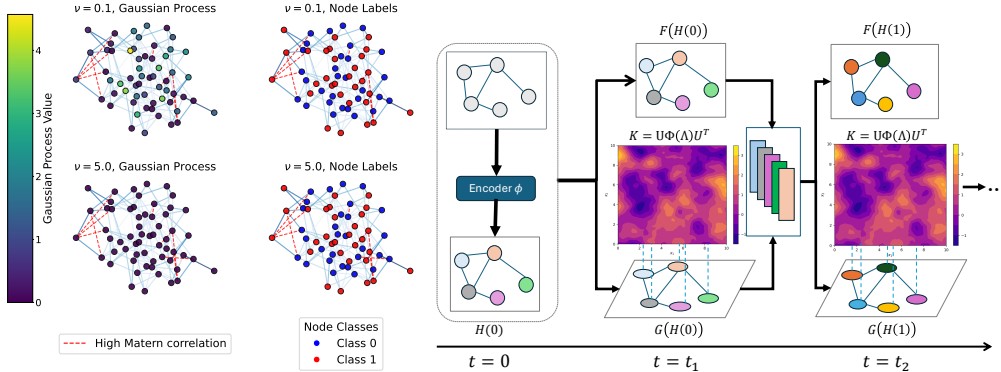

(a) Matérn Kernel on a Heterophilic Graph      (b) Structure Informed SPDE

Figure 1: (a) Graph Matérn kernel with varying degrees of smoothness $\nu$: for low $\nu$, the Gaussian random field is rough, with higher variance for each node and higher correlation between nodes. Red links are high correlation edges that do not exist in the original graph. (b) Our proposed Structure Informed SPDE (SISPDE) to incorporate spatial correlations between node uncertainty (section 3): Gaussian noises between nodes are correlated according to the Matérn Gaussian Random Field.

estimation: when labels that are close by are uninformative, GNNs that are poorly calibrated either generate incorrect predictions with high certainty or cannot capture the pattern at all [36, 35].

This potential difficulty has not been addressed by previous works but has been raised by many existing works as future directions [55, 26, 12]. In this paper, we directly address this challenge by proposing a message passing scheme on GNN that explicitly models the dependency between uncertainty of nodes through a spatially-correlated stochastic partial differential equation (SPDE) (Figure 1(b))[16]. In particular, we extend the basic framework of $Q$-Wiener process from [26], which uses the graph Laplacian to model spatial correlation, to a Matérn Gaussian process on graph [4]. With this construction, we can explicitly regulate the smoothness of the spectrum used to represent the Brownian motion on graph, creating covariance structures that can capture long-distance dependencies between uncertainties of nodes. We further provide theoretical analysis for the construction and demonstrate the existence and uniqueness of mild solution of the underlying SPDE. We conduct extensive experiments on Out-of-Distribution (OOD) Detection on 8 graph datasets with varying degrees of label informativeness to demonstrate our model's effectiveness.

The main contribution of this paper are: (1) we introduce a physics-inspired way to perform message passing on graph with spatially-correlated noise to improve uncertainty estimation, even under low label-informativeness. (2) We extend the $Q$-Wiener process on graph to a smoothness-regulated, spatially-correlated noise process and analyze its property and the theoretical implication for the underlying stochastic process on graph. (3) We provide an efficient implementation of this general framework, conduct extensive experiments on standard uncertainty estimation tasks, and provide detailed ablations and visualizations for the properties of designed covariance structure. Empirical results and thorough analyses demonstrate our model's novelty and superiority compared with previous approaches.

## 2 Background

### 2.1 Uncertainty Estimation on Graph Data

Let $G = (V, E)$ be a graph, where $V$ is the set of vertices and $E$ the set of edges, then a semi-supervised node classification problem can be formulated by (1) partitioning the graph nodes: $V = \mathcal{T} \cup \mathcal{U}$, where $v \in \mathcal{T}$ are the set of nodes with labels and $u \in \mathcal{U}$ are the set of nodes without labels, and (2) train a GNN model to predict the labels of $u \in \mathcal{U}$ using the final node features after $T$ rounds of message passing. Denote node features as $\mathbf{X}$ and $\mathbf{H}^{(T)}$ as feature after $T$ rounds of message passing, then the classification results depend on $P(y \mid \mathbf{H}^{(T)})$. Traditionally, predictive uncertainty can be measured by (a) adopting an energy-based model perspective [55, 12] on the

classifier, (b) formulating a Bayesian perspective and compute the entropy of the posterior predictive distribution [45], and (c) formulating the message passing GNN as a stochastic process and compute an expected Shannon entropy $H[\mathbb{E}[p(y \mid \mathbf{H}(T))]]$. In this paper, we investigate approach (c), which allows for a principled design of GNN architecture and explicit modeling of both aleatoric uncertainty and epistemic uncertainty through $P(\mathbf{H}(T))$ and $\mathbb{E}_{P(\mathbf{H}(T))}[H[P(y \mid \mathbf{H}(T))]]$, respectively [22, 26].

## 2.2 Label Informativeness

Traditionally for node classification tasks, graph datasets were divided into homophilic and heterophilic datasets, where similar nodes are more likely to be connected in the former and less likely to be connected in the later case [31]. While it occurred that classical GNNs perform poorly on heterophilic datasets, it has been recently discovered that fine-tuning can address the issue without relying on heterophilic-specific methods [31, 35, 36], leading to design of metrics that better characterize node classification difficulty. Among the newly proposed metrics, Label Informativeness (LI) leverages the mutual information framework and better characterizes the structural distribution of graph labels. For two nodes $i, j \in V$ and their label random variables $y_i, y_j$, LI is defined as:

$$LI := I(y_i, y_j)/H(y_i) = \frac{H(y_i) - H(y_i \mid y_j)}{H(y_i)},$$

(1)

where $I$ is mutual information and $H$ Shannon entropy. Compared with homophily, edge-wise LI can better account for classification difficulty and naturally incorporates multiple types of heterophilic patterns [35]. Using entropy and mutual information, LI provides not just a metrics of classification difficulty, but also of predictive uncertainty: intuitively, when label informativeness is low, predictive uncertainty of center node doesn't decrease even after knowing the label of the neighbors, indicating a low correlation between the center node's predictive uncertainty and the predictive uncertainty of neighbor nodes. This suggests that modeling the spatial correlation between nodes can serve as a prior for graph's label informativeness, and we argue that it is crucial to model not just uncertainty of each node, but also their correlations.

## 2.3 Graph Stochastic Partial Differential Equation

The analogy between Message Passing Neural Networks and Partial Differential Equations (PDEs) has been widely explored, with pioneering work in PDEGCN [10] and GRAND [5], which systematically defined differential operators on graph. For the node feature $\mathbf{X}$ and its corresponding node embedding $\mathbf{H}$ on graph, GRAND [5] formulates message passing using the diffusion equation:

$$\frac{\partial \mathbf{H}(t)}{\partial t} = \text{div}[G(\mathbf{H}(t), t)\nabla \mathbf{H}],$$

(2)

which closely follows the anisotropic heat equation $\frac{\partial \mathbf{X}}{\partial t} = \Delta(G(\mathbf{X}, t)\mathbf{X})$. In this formulation, $G$ is a GNN layer while $t$ is the continuous index for layers of the GNN, and the entire message passing that updates node features can be written as an integral equation. Later on, to better account for uncertainty, Graph Stochastic Neural Diffusion (GNSD) [26] expanded this formulation into the stochastic diffusion equation, a class of stochastic partial differential equations [16]:

$$d\mathbf{H}(t) = f(\mathbf{H}(t))dt + g(\mathbf{H}(t))d\mathbf{W}(t),$$

(3)

where the Brownian motion $\mathbf{W}(t)$ is defined using the spectral basis of the graph Laplacian, using the finite dimensional analogy of the Karhunen–Loève expansion:

$$\mathbf{W}(t) = \sum_{k=1}^{|V|} \sqrt{\lambda_k} \mathbf{u}_k \beta_k(t),$$

(4)

where $\{\lambda\}_{k=1}^{|V|}, \{\mathbf{u}_k\}_{k=1}^{|V|}$ are the eigenvalues and eigenvectors of the graph Laplacian, and $\{\beta_k(t)\}_{k=1}^{|V|}$ are independent zero-mean Brownian motion on each node. In particular, [26] showed that the noise process is a $Q$-Wiener process, which converges in the asymptotic regime in the $L^2$ sense, and the process has stationary increment of the form $W(t) - W(s) \sim \mathcal{N}(\mathbf{0}, (t-s)Q)$, where $Q$ is a trace class operator and defined to be the graph Laplacian in the finite case. The SPDE in the form of equation 3 suggests a message passing scheme on GNN, where noises are injected independently for each graph nodes, which better accounts for aleatoric uncertainty of the data.

## 2.4 Graph Matérn Gaussian Process and SPDE

Generally, a stochastic partial differential equation (SPDE) describes a spatial-temporal stochastic process driven by a spacetime noise process $W(x, t)$. When time is fixed, the object $W(x)$ is a Random Field, whereas if $x$ is fixed, we obtain a classical stochastic process [16]. Under regularity conditions, a Gaussian process can be formulated as the mild solution to a SPDE system [42], with its covariance structure derived from the underlying Brownian motion. In particular, when the spatial domain is a graph, [4] showed that a Gaussian process can be defined over the graph nodes, using the spectrum of the graph Laplacian matrix to form the covariance kernel. Specifically, given the spectral decomposition of the Laplacian matrix $\Delta = U\Lambda U^T$, we can define a matrix function $\Phi : \mathbb{R}^{|V| \times |V|} \to \mathbb{R}^{|V| \times |V|}$, $\Phi(\Delta) := U\Phi(\Lambda)U^T$ can be used to define a fractional diffusion operator, where $\Phi(\Lambda) = \text{diag}([\phi(\lambda_1), \cdots, \phi(\lambda_n)])$ scalar transforms the eigenvalues with scalar function $\phi$. When driven by standard Gaussian $W \sim \mathcal{N}(\mathbf{0}, \mathbf{I})$, the diffusion equation generates a spatially correlated Gaussian process. Using the general result from [54] that formulates the Matérn Gaussian process as the mild solution of class of diffusion type static SPDEs, [4] proposed the family of Matérn kernel and the associated Matérn Gaussian process on graph:

$$\Phi(\Delta) = \left(\frac{2\nu}{\kappa^2} + \Delta\right)^{\nu/2}, \quad \mathbf{f} \sim \mathcal{N}\left(\mathbf{0}, \left(\frac{2\nu}{\kappa^2} + \Delta\right)^{-\nu}\right), \tag{5}$$

where $\nu, \kappa > 0$ are spatial scaling parameters and $\mathbf{f} : V \to \mathbb{R}^d$ can be seen as a random field defined on the graph domain. This formulation provides a framework to model the smoothness of spatial correlation structure for noise on graph. When $\nu$ is large, the noise process is smooth, with low covariance overall, whereas if $\nu$ is small, the noise process is rough, resulting in long distance dependencies. A visualization of Matérn kernel on a synthetic graph can be found in Figure 1(a).

## 3 Method

In this section, we first design a general spacetime noise process on graph using Gaussian Random Field and Brownian motion. Based on this general design, we propose a spectral formulation of the noise process and discuss the design of the spatial covariance kernel. We formalize the findings in four theorems and provide the detailed proofs in Appendices. We then provide an efficient graph neural ODE implementation and a learning framework to handle uncertainty estimation.

### 3.1 $\Phi$-Wiener Process on Graph

In GNSD[26], the authors adopted a spectral perspective and proposed to represent the spacetime white noise using the graph Laplacian. Although the construction is a valid $Q$-Wiener process, it lacks a rich spatial structure: the noise process defined in GNSD corresponds to a Gaussian Random Field whose spatial covariance kernel is the Laplacian Matrix.

**Proposition 1.** *Let $G = (V, E)$ be a graph and $i, j \in V$, then the Q-Wiener process defined in Equation 4 results in a spatial covariance structure of $Cov(W_i(t), W_j(t)) = \Delta_{ij}t$.*

While this result provides some degrees of spatial correlation between noise on graph nodes, it doesn't provide a mechanism to control for kernel smoothness, an important prior when dealing with label distributions, limiting the kind of uncertainty that may be encoded in the graph structure and potentially harming the performance of uncertainty estimation on graph. To overcome this issue, we propose a more flexible way to encode graph structure by directly modeling the spatial correlation pattern as a covariance kernel. To facilitate this construction, we propose the following definition:

**Definition 1.** *(Graph Gaussian Random Field) Let $G = (V, E)$ be an undirected graph and $\Delta$ its graph Laplacian, then a Graph Gaussian Random Field is a multivariate Gaussian random variable with its positive definite covariance matrix $K = U\Phi(\Lambda)U^T$, where $\Delta = U\Lambda U^T$ is the spectral decomposition of the graph Laplacian, and $\Phi$ represents a matrix-valued function that applies a scalar function to each element of the diagonal matrix $\Lambda$.*

This is a Gaussian random variable that directly models the correlation structure between graph nodes. In [4], when $\phi$ equals the transformation in equation 5, a Matérn Gaussian process can be obtained. Denote $\Phi$ the matrix functional form of $\phi$, which corresponds to element-wise transform of matrix

element using $\phi$, we propose the following definition for spatially-correlated Wiener process on graph induced by the function $\Phi$:

**Definition 2.** *($\Phi$-Wiener process on graph). Let $G = (V, E)$ be an undirected graph and $\Delta$ its graph Laplacian. Let $\{\mathbf{u}_k\}_{i=1}^{|V|}, \{\lambda_k\}_{k=1}^{|V|}$ be its eigenvectors and eigenvalues, respectively, and let $\phi : \mathbb{R} \to \mathbb{R}$ be a scalar function with $\Phi$ its matrix functional form, then a $\Phi-$Wiener process is defined by the (truncated) Karhunen-Loève expansion:*

$$W(t) = \sum_{k=1}^{|V|} \sqrt{\phi(\lambda_k)}\mathbf{u}_k\beta_k(t), \tag{6}$$

*where $\{\beta_k(t)\}_{k=1}^{|V|}$ are independent Brownian motions on each graph node.*

Compared with the construction in [26] (Equation 4), we explicitly introduce a covariance structure between nodes using the function $\phi$. This construction has the following property:

**Theorem 1.** *Let $G = (V, E)$, and $\Delta$ its graph Laplacian with $\Delta = U\Lambda U^T$, then for the construction in definition 2, if the matrix $K = U\Phi(\Lambda)U^T$ is positive definite, then the spatial-temporal covariance of noise process on two nodes $j, k$ can be written as $Cov(W_i(t), W_j(s)) = (t \wedge s)K_{ij}$, where $(t \wedge s) = min(t, s)$, and $W(t) - W(s) \sim \mathcal{N}(\mathbf{0}, (t - s)K)$ for $t > s$.*

*Sketch Proof.* The result follows from the orthogonality of the eigenvectors of the graph Laplacian and the fact that Wiener processes are zero-centered with correlation structure of $Cov(W(t), W(s)) = min(t, s)$ and $W(t) - W(s) \sim \mathcal{N}(0, t - s)$ for $t > s$. $\square$

In the asymptotic regime where $|V| \to \infty$, the process generalizes to be a space-time noise process with the following property (proof in Appendix A.5):

**Theorem 2.** *Let $\mathcal{H}$ be a separable Hilbert space and $(\Omega, \mathcal{F}, \mathcal{F}_t, \mathbb{P})$ a filtered probability space. Let $Q'$ be a trace-class operator in $\mathcal{H}$ and $\Phi$ a function operator that scalar-transforms the eigenvalues $\{\lambda_i\}$ of $Q'$, then if $\sum_{i=1}^{\infty} \sqrt{\phi(\lambda_i)} < \infty$, the induced $\Phi$-Wiener process is a $Q$-Wiener process with the trace class operator $Q = \Phi(Q')$. Moreover, the transformation for Matérn kernel induces a $Q$-Wiener process when $\nu > d$ for a random field taking value in compact $U \subset \mathbb{R}^d$.*

The covariance structure derived in theorem 1 implies that the $\Phi$-Wiener process constructed in this way is more expressive than the $Q$-Wiener process constructed in [26], and the condition outlined in theorem 2 provides the framework to reason about the solution of SPDE driven by such a noise process, which we highlight in theorem 3 in the section below.

## 3.2 Structure Informed Graph SPDE

Aside from the theoretical construction, we propose that the design of function $\Phi$ should depend on the underlying graph structure and label informativeness. In the context of graph node classification, various previous works [55, 26] assumed that nodes that are connected to each other should have similar uncertainty. However, for graphs with low label informativeness, knowing the labels of neighbors does not decrease of entropy for the center node[1, 31]. This suggests that far-apart nodes can exhibit similar uncertainty patterns, while the magnitude of noise between neighboring nodes can vary greatly. In the language of Graph Gaussian Random Field 1, this means a slowly decaying covariance between two nodes that are far apart. To incorporate this insight, we propose to carefully design $\phi$ using the Matérn Kernel in Equation 5. In this case, the choice of $\nu$ is crucial, for high values of $\nu$ indicate a smooth and hence short-range dependency, whereas low values of $\nu$ a rough and hence long range dependency. The SPDE driven by the Gaussian process with the Matérn kernel can then be used as a message passing scheme to address the distribution of uncertainty in both cases. We summarize the theoretical property of this SPDE in the following theorem:

**Theorem 3.** *Let $\mathcal{H}$ be a separable Hilbert space, and $(\Omega, \mathcal{F}, \mathcal{F}_t, \mathbb{P})$ be a filtered probability space and $W(x, t)$ be a $\mathcal{F}_t$-adapted space-time stochastic process whose spatial covariance kernel is given by the Matérn kernel on a closed and bounded domain $D \subset \mathbb{R}^d$, then if $v > d$ and $\mathbf{H}(0)$ is square-integrable and $\mathcal{F}_0$-measurable, the SPDE of the form:*

$$d\mathbf{H}(t) = \Big(\mathcal{L}\mathbf{H}(t) + F(\mathbf{H}(t))\Big)dt + G(\mathbf{H(t)})dW(t) \tag{7}$$

*admits a unique mild solution:*

$$\mathbf{H}(t) = \exp(t\mathcal{L})\mathbf{H}(0) + \int_0^t \exp((t-s)\mathcal{L})F(\mathbf{H}(s))ds + \int_0^t \exp((t-s)\mathcal{L})G(\mathbf{H}(s))dW(s), \quad (8)$$

*where $\mathcal{L}$ is a bounded linear operator generating a semigroup $\exp(t\mathcal{L})$ and $F, G$ are global Lipschitz functions.*

This theorem holds generally even when the noise process $dW_t$ is not a $Q$-Wiener process [Chapter 6 in [16] [41]]. In section 4, we provide empirical results for the choice of $\nu$. In the section below, we provide an alternative formulation using the mollifier proposed in [41, 11] and present our neural architecture incorporating the structure aware graph SPDE.

**Remark 1.** *(Expressiveness): Our proposed Structure Informed Graph SPDE is more expressive than the PDE formulation in GRAND [5] because of the additional noise term. It is also more expressive than GNSD [26], which corresponds to the special case where $\Phi$ is the identity transformation. Compared to GREAD [7], which uses deterministic forcing, our model uses stochastic forcing.*

### 3.3 Practical Implementation: Graph ODE with Random Forcing

For the SPDE of the form 7, its solution can be written with a mollifier $W^\epsilon = \psi^\epsilon * W$, where $\psi$ is a smooth test function and $*$ is the convolution operator [11, 41, 20]. The solution then can be written as a randomly forced PDE:

$$\mathbf{H}(t) = \exp(t\mathcal{L})\mathbf{H}(0) + \int_0^t \exp\Big((t-s)\mathcal{L}\Big)\Big(F(\mathbf{H}(s)) + G(\mathbf{H}(s))\xi_s\Big)ds, \quad (9)$$

where $\xi_t = W_t^\epsilon$ is a noise process that serves as mollification of the underlying Wiener process. From the classical theory of Wong-Zakai approximation [49], this construction ensures that for SDEs, the sequence of random ODEs driven by the mollification of Brownian motion converges in probability to a limiting process independent from the mollifier. In [41], it is defined as a colored noise when $W_t$ is a $Q$-Wiener process and white noise if $W_t$ is a cylindrical Wiener process. The corresponding neural architecture consists of modeling the semigroup $\exp(t\mathcal{L})$ using a convolution kernel, as done in [41], which in our case is a Graph Neural Network [24], and the functions $F, G$ are MLPs. The first step therefore performs graph convolution and the resulting architecture is a Graph Neural ODE [37] with a spatially correlated, temporally stationary Gaussian process injected at each time step:

$$\mathbf{H}(t) = \text{ODESolve}(\text{GNN}_\theta(\mathbf{H}(0)), \Psi_{\theta,\xi}, [0, t]), \quad (10)$$

where the integrand has the form:

$$\Psi_{\theta,\xi}(\mathbf{H}(t)) = \text{GNN}_\theta(F_\theta(\mathbf{H}(t)) + G_\theta(\mathbf{H}(t))\xi_t) \quad (11)$$

$$\xi_t \sim \mathcal{N}\left(\mathbf{0}, t\Big(\frac{2\nu}{\kappa^2} + \Delta\Big)^{-\nu}\right) \quad (12)$$

**Approximating the Matérn Kernel**: We simulate the Multivariate Gaussian distribution $\xi_t$ using the Cholesky decomposition on the Matérn covariance matrix on graph [4]. For graphs with a large number of nodes, performing an eigendecomposition on the Laplacian matrix is prohibitively expensive ($\mathcal{O}(|V|^3)$), so we adopt a Chebyshev polynomial approximation of the Matérn kernel [8], in which case we avoid Cholesky decomposition and directly use the following expansion to approximate the kernel matrix:

$$\Big(\frac{2\nu}{\kappa^2} + \Delta\Big)^{-\nu} \approx \sum_{k=0}^m c_k T_k(\tilde{\Delta})\beta_i, \quad (13)$$

where $\{T_k\}_{k=1}^m$ are Chebyshev polynomials up to degree $m$ and $\{c_k\}_{k=1}^m$ are Chebyshev coefficients determined by $\phi(x) = (2\nu/\kappa^2 + x)^{-\nu}$, and $\tilde{\Delta}$ is the rescaled graph Laplacian, resulting in $\mathcal{O}(m|E|)$ complexity. We summarize the theoretical property of this approximation in the theorem below:

**Theorem 4** (Chebyshev Approximation of Matérn kernel). *Let $G = (V, E)$ be a graph, and let the Matérn kernel on graph be defined in equation 5, then the Chebyshev approximation in equation 13 results in an error bound of $\mathcal{O}\left(\left(\frac{\kappa^2 d_{max}}{16\nu + \kappa^2 d_{max}}\right)^m\right)$, where $m$ is the degree of the Chebyshev polynomial and $d_{max}$ is the maximum degree of the graph.*

We provide the technical proof for this theorem in Appendix B. Intuitively, for large $\nu$, which indicates a smoother covariance structure, the error rate decays exponentially with respect to the degree of the polynomials. For model training, we adopt the distributional uncertainty loss used in [26], which relies on the predictive distribution of the final state $\mathbf{H}(T)$:

$$L(y, \hat{y}) = \mathbb{E}_{P(\mathbf{H}(T))}[H(P(y \mid \mathbf{X}, G), P(\hat{y} \mid \mathbf{H}(T)))], \tag{14}$$

where $H$ is the Shannon entropy. In the experiment section, we investigate the choice of parameters $\nu$, which controls the roughness of the Gaussian Random Field.

## 4 Experiments

In this section, we conduct extensive experiments to demonstrate the capacity of our model: in section 4.1, we demonstrate model performance on Out-of-distribution (OOD) detection on semi-supervised node classification task under the influence of label, feature, and structure shifts, on both low LI and high LI datasets. In section 4.2, we examine the impact of kernel smoothness parameter $\nu$ on model performance with respect to label informativeness. In section 4.3, we provide a graph rewiring perspective on stochastic message passing. We provide comprehensive comparisons with previous works, especially with GNSD[26] for OOD detection tasks. Additional experiment details for hyperparameter search, visualizations, and results can be found in Appendix C.

### 4.1 OOD Detection

We evaluate our model's capacity to model uncertainty for graph data using the task of Out-of-Distribution Detection (OOD) for node classification. This task involves training the models using node classification task, while reporting the OOD Detection performance at test time. To highlight the effectiveness of modeling the spatial patterns of noise, we experiment on graph datasets with varying label informativeness. This includes heterophilous / low LI datasets [36] (Roman Empire, Amazon-Ratings, Minesweeper, Tolokers, and Questions) and high LI datasets (Cora [30] Citeseer[43], and Pubmed [34]). Details of dataset LIs can be found in Appendix C. For each dataset, we perform following three types of distribution shifts, similar to previous works [26, 55]:

- **Label-Leave-out**: We treat a subset of class labels as OOD and leave them out of training. For the common benchmarks, we follow the approach in [26] and leave the class with partial labels as the OOD class. For the heterophilous datasets, we use the last class label as the OOD class for Minesweeper, Tolokers, and Questions, and more labels for Roman Empire and Amazon Ratings for more balanced distribution. We list the detailed label assignments for each dataset in Appendix C.

- **Structural Perturbation**: we follow [55] by using original graph as in-distribution data and applying stochastic block model to randomly generate a graph for OOD data.

- **Feature Perturbation**: we follow [26, 55] and use the original graph as in-distribution data while perturbing a subset of nodes' features with Gaussian noise to generate OOD samples.

**Baseline Models**: we compare our model with non-graph models specifically designed for OOD detection (ODIN [25], MSP [17], Mahalanobis [23] OE[18]), GNN-based OOD models (GCN[21], GNNSafe [55], GOLD [52], GEBM [12]), and stochastic message passing based model (GNSD [26]). We also add a special case of SISPDE-RBF to compare the Matérn kernel agains the smooth RBF kernel. We focus on the case where the OOD labels are not available during training, making our approach more general than the scenarios in [55, 45], which rely on existing OOD labels to help with the OOD detection task. In this OOD scenario, the model sees the normal and learns what the normal looks like at training time and detect the abnormal at test time from normal and abnormal samples. Some previous works have explored the case of OOD samples exposure during training time, which allows them to further the gap between IND and OOD sample's entropy or energy function, but in our case we choose to handle the more general scenario of no OOD exposures during training. For more recent datasets that previous works have never been run on, we conduct hyperparameter search for some of the more recent baseline models using the search space provided by them.

**Evaluation Metrics**: In order to study the capacity of the model to detect OOD samples, we formulate the problem as a binary classification task and quantify the performance using detection accuracy

(DET ACC), AUC, and FPR95, where larger values for DET ACC and AUC means better OOD detection performance, whereas smaller value of FPR95 indicates better OOD detection performance. In Table 1, we report the results of baseline models on all the datasets, with each metrics reported as mean ± std over 5 runs with different seeds. We use green and orange to signify the best performance and the runner-up performance.

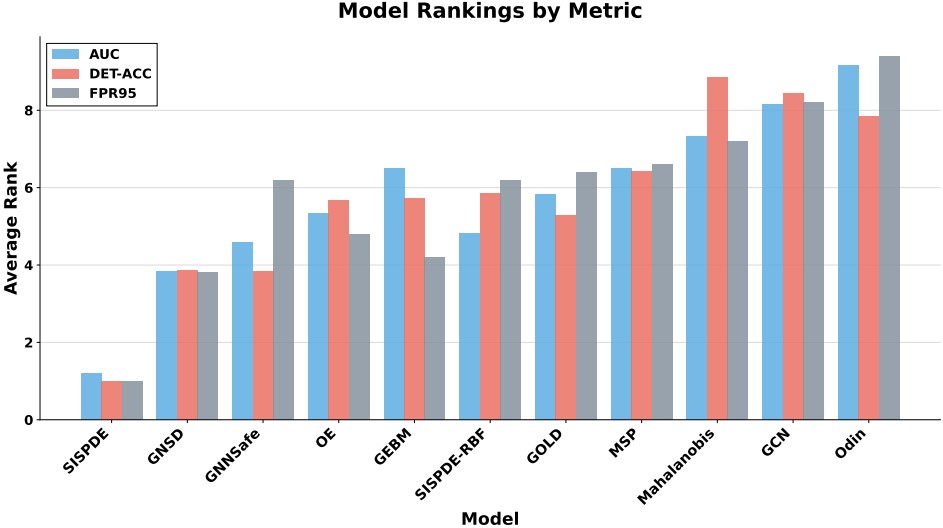

(a) Average Rank of Model for each metrics

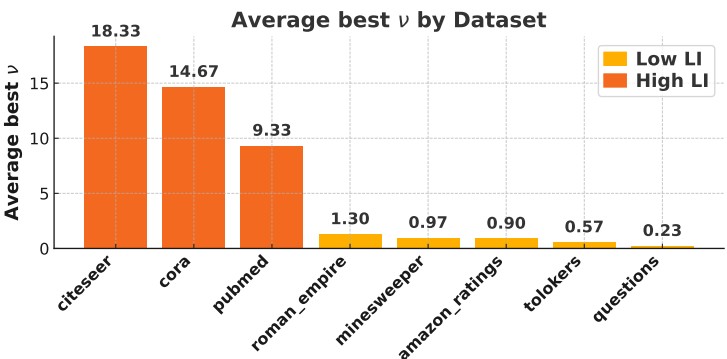

(b) Best value of $\nu$ for each dataset

Figure 2: (a) Average rank of metrics for model on all graph datasets in Table 1; (b) Comparison of smoothness parameter $\nu$ on each dataset. Here we plot the average $\nu$ over the three cases (label, structure, feature) for each dataset.

**Summary of Results**: The results of the experiments can be found in Table 1. Of the 72 results, our model (SISPDE) achieves the best result or the second best in 71 cases (98.6% of all experiments). In particular, it performs the best in 50 cases (69.4% of all) and the second best in 21 cases (29.2% of all), demonstrating our model's consistent and superior performance not just on homophilic datasets with high label informativeness (Cora, Pubmed, Citeseer) but also on heterophilic datasets that have low label informativeness (Roman Empire, Minesweeper, Questions, Tolokers, Amazon Ratings). Compared to GNSD, which assumes a smooth covariance kernel, our model achieves better performance in 26 of the 27 cases (96.2% of all) for data sets with high LI and 45 of the 45 (100% of all) for datasets with low LI, demonstrating the validity of our assumption on the importance of modeling spatial correlation. We present the average rank of each model in Figure 2(a). Across all datasets and metrics, our model's average rank is the best, which holds true also for low LI datasets, as demonstrated in Figure 3.

## 4.2 Kernel Smoothness and Label Informativeness

In this section we examine the impact of the kernel smoothness parameter $\nu$ in the Matérn kernel on OOD Detection on graphs with varying label informativeness. Here we state our observations:

**Observation 1: Lower kernel smoothness benefits learning under low label informativeness**: In Figure 2 (b), we demonstrate the best $\nu$ for low label informative datasets to be significantly smaller than datasets with high label informativeness. We reason that a smooth spatial covariance structure (large $\nu$) results in short-range uncertainty dependencies whereas a rough covariance structure (small $\nu$) results in a long-range uncertainty dependency, which should benefit the case for low informative labels. We also provide the case where $\nu \to \infty$, where the Matérn kernel turns into the RBF kernel [4]. In table 1. The results demonstrate that in general, flexible choice of kernel smoothness outperforms the smooth behavior provided by the RBF kernel: the RBF kernel results consistently perform poorly on the low LI datasets.

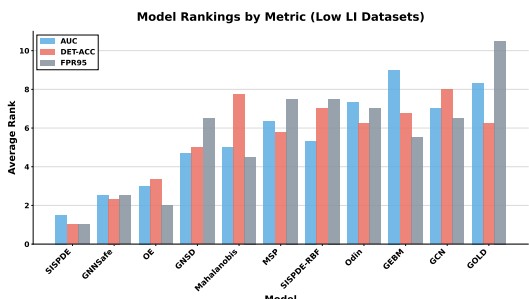

Figure 3: Average rank for low LI datasets. Some baselines perform better than GNN models.

**Observation 2: Traditional models without graph structure can perform better on low LI datasets**: In Figure 3, we provide the average rank for all the models on low label informative datasets and observe that models without using graph structure (OE, ODIN) can perform better than previous state-of-art graph models (GNNSafe, GNSD) when the graph model does not particularly consider the problem of low label informativeness. Our SISPDE instead consistently outperforms all the baselines by explicitly modeling the spatial correlation of uncertainty.

## 4.3 Covariance Kernel as Implicit Dynamic Rewiring

In this section we provide an alternative insight on why covariance kernel design helps with learning on low LI graphs by performing the following experiment: for a graph and its Matérn covariance kernel, we set a threshold value based on percentile of covariance distribution, then prune the low correlation edges and insert new edges that have high correlations. After rewiring, the label informativeness of roman empire increases by a factor of $2.21$, tolokers by $2.94$, minesweeper by $2.90$, and Questions by $1.84$. We then use these rewired graphs as the new input to the simple GCN model and compare the performances. Comparison of OOD detection results on label leave out scenario can be found in Table 2, where GCN's OOD detection performance improves significantly after rewiring. This suggests that spatial covariance structures can effectively serve as a prior for label informativeness.

## 5 Conclusion and Future Work

In this paper, we proposed a novel, systematic, and rigorous framework to perform message passing on graph motivated by stochastic partial differential equations driven by spatially correlated Gaussian process. We demonstrate the soundness and superiority of the framework on uncertainty estimation task for graph datasets with varying label informativeness and provide insights on the mechanism by analyzing kernel smoothness and graph rewiring. There are a number of promising and intriguing directions to work on: (a) Adapt a systematic Bayesian perspective and use the spatially-correlated SDE as a prior to extend the current framework for uncertainty estimation. (b) Explore the more general case of nonlinear dependency between uncertainty (such as mutual information). (c) Exploring covariance structures for graph substructures rather than nodes. (d) Further explore a larger class of SPDE models, such as those with both a deterministic and stochastic forcing term or with fractional brownian motions. (e) Improve the scalability of the current approach on larger graphs. (f) Explore application to other tasks such as modeling stochastic spatial-temporal dynamical systems.

| | Model | Label | | | Structure | | | Feature | | |
|---|---|---|---|---|---|---|---|---|---|---|
| | | AUC↑ | DET-ACC↑ | FPR95↓ | AUC↑ | DET-ACC↑ | FPR95↓ | AUC↑ | DET-ACC↑ | FPR95↓ |
| Cora | GCN | 83.91 ± 1.46 | 76.53 ± 0.92 | 64.55 ± 0.97 | 68.49 ± 3.34 | 59.45 ± 3.60 | 84.77 ± 2.43 | 59.45 ± 1.30 | 52.86 ± 2.92 | 88.64 ± 2.17 |
| | Odin | 47.61 ± 3.09 | 50.40 ± 0.25 | 95.33 ± 0.97 | 67.40 ± 1.43 | 62.59 ± 0.72 | 92.79 ± 1.67 | 61.91 ± 4.82 | 58.86 ± 4.08 | 91.84 ± 1.50 |
| | Mahalanobis | 74.97 ± 1.49 | 65.61 ± 1.45 | 64.96 ± 3.66 | 53.50 ± 2.76 | 53.57 ± 0.56 | 96.94 ± 0.52 | 58.05 ± 0.87 | 55.67 ± 0.81 | 92.41 ± 0.36 |
| | GNNSafe | 90.74 ± 0.49 | 85.52 ± 1.39 | 46.86 ± 4.19 | 86.84 ± 0.31 | 83.10 ± 0.21 | 81.38 ± 3.30 | 89.89 ± 1.06 | 86.85 ± 0.65 | 79.34 ± 11.73 |
| | GNSD | 95.75 ± 0.31 | 89.85 ± 0.74 | 19.76 ± 0.81 | 91.53 ± 0.95 | 86.52 ± 1.07 | 26.23 ± 3.46 | 97.48 ± 0.21 | 93.49 ± 0.24 | 7.66 ± 0.47 |
| | MSP | 91.43 ± 0.23 | 84.54 ± 0.61 | 36.96 ± 0.34 | 73.15 ± 0.65 | 68.89 ± 0.63 | 84.28 ± 1.28 | 85.40 ± 0.91 | 78.13 ± 0.88 | 62.96 ± 2.23 |
| | OE | 90.24 ± 0.47 | 82.76 ± 0.69 | 42.07 ± 2.04 | 70.59 ± 0.68 | 66.79 ± 0.58 | 90.33 ± 1.92 | 82.73 ± 1.03 | 75.25 ± 0.64 | 74.89 ± 3.79 |
| | GOLD | 93.49 ± 1.85 | 85.94 ± 2.00 | 36.82 ± 15.58 | 95.94 ± 0.47 | 88.53 ± 4.87 | 19.72 ± 3.02 | 96.72 ± 0.51 | 92.45 ± 0.63 | 12.68 ± 3.85 |
| | GEBM | 90.50 ± 11.23 | 79.90 ± 10.15 | 30.80 ± 13.73 | 78.50 ± 9.95 | 77.70 ± 5.51 | 51.00 ± 8.29 | 75.00 ± 7.81 | 73.20 ± 2.73 | 61.00 ± 2.79 |
| | SISPDE-RBF | 95.22 ± 0.54 | 88.90 ± 0.75 | 23.67 ± 5.62 | 83.55 ± 2.24 | 77.93 ± 2.61 | 77.63 ± 6.86 | 91.38 ± 0.84 | 85.30 ± 1.15 | 61.57 ± 8.03 |
| | **SISPDE** | 96.85 ± 0.68 | 91.45 ± 0.13 | 10.87 ± 2.96 | 93.05 ± 1.36 | 88.56 ± 0.74 | 18.94 ± 0.85 | 97.89 ± 0.29 | 94.34 ± 0.55 | 6.17 ± 0.93 |
| Pubmed | GCN | 58.23 ± 0.42 | 56.85 ± 0.66 | 93.17 ± 0.49 | 88.86 ± 0.54 | 82.11 ± 0.75 | 56.71 ± 3.04 | 81.24 ± 0.28 | 77.02 ± 0.22 | 84.72 ± 0.29 |
| | Odin | 52.17 ± 2.58 | 52.54 ± 1.60 | 95.66 ± 0.52 | 78.27 ± 0.70 | 76.58 ± 0.79 | 99.91 ± 0.06 | 57.73 ± 2.35 | 56.56 ± 1.89 | 94.50 ± 0.42 |
| | Mahalanobis | 72.94 ± 3.43 | 55.41 ± 0.29 | 66.34 ± 4.92 | 63.41 ± 0.77 | 55.03 ± 0.20 | 81.23 ± 2.57 | 70.14 ± 2.21 | 55.35 ± 0.30 | 75.15 ± 2.50 |
| | GNNSafe | 76.34 ± 2.33 | 69.60 ± 2.14 | 72.23 ± 5.83 | 87.12 ± 0.77 | 84.15 ± 1.74 | 92.15 ± 1.36 | 86.88 ± 2.05 | 83.68 ± 1.32 | 93.38 ± 8.46 |
| | GNSD | 79.91 ± 3.69 | 73.99 ± 3.65 | 68.62 ± 6.39 | 97.59 ± 0.14 | 94.42 ± 0.29 | 5.82 ± 0.66 | 98.57 ± 0.50 | 94.55 ± 1.17 | 5.76 ± 1.87 |
| | MSP | 71.45 ± 0.69 | 66.58 ± 0.50 | 72.50 ± 3.33 | 73.91 ± 0.67 | 68.32 ± 0.52 | 80.73 ± 0.43 | 83.20 ± 0.76 | 75.14 ± 0.26 | 65.22 ± 1.02 |
| | OE | 70.58 ± 5.25 | 65.32 ± 4.58 | 68.34 ± 9.42 | 76.41 ± 1.09 | 70.66 ± 0.74 | 82.29 ± 2.29 | 84.59 ± 1.35 | 77.37 ± 1.39 | 66.98 ± 5.55 |
| | GOLD | 68.62 ± 2.98 | 62.81 ± 2.42 | 83.73 ± 4.16 | 96.07 ± 1.56 | 89.46 ± 3.08 | 21.21 ± 16.25 | 98.24 ± 0.49 | 90.81 ± 2.25 | 7.83 ± 3.01 |
| | GEBM | 79.50 ± 2.36 | 64.60 ± 0.79 | 64.30 ± 7.16 | 94.70 ± 1.70 | 91.40 ± 1.21 | 12.70 ± 3.28 | 90.30 ± 2.48 | 90.30 ± 2.48 | 14.90 ± 9.06 |
| | SISPDE-RBF | 79.16 ± 5.82 | 73.59 ± 4.62 | 62.92 ± 12.07 | 95.19 ± 1.91 | 89.56 ± 1.76 | 23.17 ± 20.22 | 90.97 ± 5.79 | 85.61 ± 5.64 | 51.30 ± 32.35 |
| | **SISPDE** | 81.50 ± 2.74 | 75.61 ± 1.82 | 56.96 ± 8.40 | 97.29 ± 0.33 | 94.56 ± 0.70 | 5.36 ± 0.87 | 99.13 ± 0.22 | 96.04 ± 0.43 | 2.58 ± 1.37 |
| Citeseer | GCN | 62.49 ± 3.16 | 59.80 ± 2.79 | 88.38 ± 0.94 | 60.93 ± 2.01 | 59.48 ± 1.82 | 90.76 ± 0.73 | 57.38 ± 2.03 | 53.18 ± 2.13 | 90.63 ± 0.96 |
| | Odin | 53.14 ± 2.55 | 61.53 ± 1.84 | 95.96 ± 0.37 | 59.23 ± 0.97 | 69.65 ± 0.88 | 94.07 ± 0.97 | 55.15 ± 1.94 | 69.02 ± 1.40 | 93.08 ± 0.94 |
| | Mahalanobis | 65.61 ± 2.61 | 61.70 ± 1.89 | 83.40 ± 1.89 | 56.25 ± 0.66 | 55.74 ± 0.40 | 94.40 ± 0.81 | 60.25 ± 1.63 | 57.67 ± 0.93 | 91.56 ± 0.65 |
| | GNNSafe | 76.53 ± 0.89 | 71.68 ± 0.30 | 74.77 ± 1.80 | 73.52 ± 1.05 | 69.98 ± 0.62 | 82.86 ± 0.70 | 72.22 ± 0.79 | 70.81 ± 0.45 | 97.31 ± 0.25 |
| | GNSD | 82.41 ± 0.62 | 76.66 ± 0.26 | 55.34 ± 1.86 | 82.25 ± 1.30 | 76.35 ± 1.13 | 66.76 ± 4.92 | 91.54 ± 0.70 | 84.41 ± 0.83 | 35.20 ± 4.76 |
| | MSP | 79.94 ± 0.68 | 73.55 ± 0.77 | 64.26 ± 1.37 | 65.63 ± 0.81 | 62.13 ± 0.36 | 84.73 ± 0.52 | 77.78 ± 0.60 | 70.03 ± 0.26 | 70.19 ± 2.28 |
| | OE | 74.19 ± 1.44 | 68.05 ± 1.19 | 80.42 ± 0.44 | 58.41 ± 1.21 | 57.19 ± 0.76 | 91.65 ± 0.95 | 83.07 ± 0.14 | 75.95 ± 0.12 | 75.48 ± 0.21 |
| | GOLD | 81.97 ± 0.62 | 75.44 ± 0.98 | 54.43 ± 4.69 | 85.44 ± 4.76 | 75.31 ± 10.92 | 78.04 ± 12.58 | 88.86 ± 1.31 | 79.10 ± 2.67 | 62.10 ± 7.07 |
| | GEBM | 94.20 ± 4.52 | 88.60 ± 5.32 | 26.90 ± 13.24 | 80.80 ± 8.00 | 75.90 ± 5.66 | 77.40 ± 3.08 | 82.80 ± 2.55 | 75.40 ± 1.39 | 57.30 ± 4.21 |
| | SISPDE-RBF | 80.26 ± 2.31 | 73.83 ± 2.52 | 62.95 ± 5.88 | 73.64 ± 2.74 | 69.78 ± 1.91 | 88.50 ± 8.15 | 83.45 ± 3.57 | 77.55 ± 3.26 | 80.33 ± 12.48 |
| | **SISPDE** | 85.43 ± 0.80 | 78.17 ± 0.67 | 53.74 ± 2.70 | 84.39 ± 1.67 | 79.56 ± 2.23 | 46.80 ± 5.55 | 92.05 ± 2.85 | 85.96 ± 2.23 | 25.23 ± 3.15 |
| Roman Empire | GCN | 53.15 ± 0.35 | 53.12 ± 0.37 | 92.49 ± 0.79 | 60.25 ± 0.28 | 58.11 ± 0.26 | 93.70 ± 0.14 | 51.17 ± 0.22 | 50.68 ± 0.24 | 94.47 ± 0.14 |
| | Odin | 57.81 ± 3.70 | 56.82 ± 2.47 | 92.95 ± 2.24 | 55.87 ± 2.14 | 56.08 ± 0.53 | 89.18 ± 3.39 | 61.49 ± 1.85 | 64.48 ± 3.76 | 88.66 ± 3.11 |
| | Mahalanobis | 51.67 ± 1.70 | 55.79 ± 1.41 | 95.91 ± 1.35 | 58.12 ± 0.90 | 54.84 ± 0.16 | 88.00 ± 0.46 | 57.70 ± 1.70 | 59.05 ± 0.14 | 85.15 ± 0.60 |
| | GNNSafe | 60.85 ± 0.40 | 58.10 ± 0.37 | 86.20 ± 0.57 | 75.73 ± 1.17 | 74.26 ± 1.73 | 46.28 ± 3.78 | 53.59 ± 1.58 | 56.43 ± 1.59 | 89.15 ± 4.42 |
| | GNSD | 56.40 ± 1.57 | 55.68 ± 0.71 | 93.99 ± 0.47 | 77.87 ± 2.71 | 72.16 ± 1.88 | 45.10 ± 0.64 | 62.14 ± 0.40 | 59.20 ± 0.47 | 91.21 ± 0.23 |
| | MSP | 66.69 ± 0.40 | 62.41 ± 0.43 | 86.35 ± 0.33 | 69.15 ± 0.40 | 64.86 ± 0.97 | 69.93 ± 0.95 | 57.50 ± 0.42 | 55.09 ± 0.27 | 91.98 ± 0.27 |
| | OE | 53.90 ± 0.54 | 55.22 ± 0.56 | 93.42 ± 5.84 | 71.82 ± 0.59 | 68.60 ± 0.20 | 68.31 ± 1.09 | 57.73 ± 0.23 | 55.87 ± 0.12 | 91.83 ± 0.22 |
| | GOLD | 49.09 ± 1.65 | 50.33 ± 0.59 | 95.58 ± 0.58 | 72.48 ± 4.88 | 73.46 ± 1.06 | 48.89 ± 2.68 | 49.92 ± 8.55 | 51.88 ± 2.08 | 95.67 ± 2.77 |
| | GEBM | 57.80 ± 2.90 | 54.60 ± 2.33 | 89.60 ± 0.35 | 49.60 ± 0.38 | 53.70 ± 0.21 | 99.90 ± 0.04 | 60.80 ± 1.08 | 57.90 ± 0.82 | 87.90 ± 0.98 |
| | SISPDE-RBF | 51.76 ± 0.57 | 51.63 ± 0.29 | 93.24 ± 0.29 | 56.58 ± 0.68 | 57.80 ± 0.55 | 81.94 ± 0.92 | 51.50 ± 0.26 | 51.34 ± 0.15 | 94.13 ± 0.08 |
| | **SISPDE** | 69.49 ± 0.30 | 65.63 ± 1.00 | 87.34 ± 1.56 | 96.12 ± 3.56 | 90.72 ± 4.38 | 13.43 ± 9.44 | 63.4 ± 0.81 | 61.35 ± 0.89 | 90.05 ± 2.85 |
| Mine Sweeper | GCN | 53.97 ± 3.22 | 53.20 ± 2.90 | 92.80 ± 2.62 | 55.91 ± 1.55 | 57.72 ± 3.56 | 82.13 ± 8.10 | 54.68 ± 1.94 | 57.64 ± 0.54 | 95.05 ± 1.28 |
| | Odin | 51.09 ± 2.62 | 51.91 ± 1.57 | 93.34 ± 1.14 | 50.73 ± 1.50 | 54.41 ± 0.31 | 93.30 ± 3.14 | 47.65 ± 1.11 | 55.15 ± 0.26 | 98.03 ± 0.26 |
| | Mahalanobis | 51.08 ± 4.22 | 52.01 ± 1.78 | 92.96 ± 1.76 | 39.19 ± 6.48 | 50.46 ± 0.41 | 95.95 ± 2.86 | 50.61 ± 1.96 | 51.42 ± 0.51 | 92.65 ± 1.54 |
| | GNNSafe | 57.46 ± 1.00 | 55.96 ± 0.85 | 91.98 ± 0.59 | 96.34 ± 0.70 | 93.06 ± 0.14 | 10.36 ± 3.43 | 93.41 ± 0.35 | 86.15 ± 0.71 | 25.60 ± 0.98 |
| | GNSD | 55.20 ± 0.69 | 54.37 ± 0.52 | 92.18 ± 0.65 | 53.19 ± 1.80 | 58.96 ± 0.13 | 95.92 ± 0.73 | 58.61 ± 3.36 | 59.43 ± 1.29 | 94.64 ± 1.40 |
| | MSP | 62.92 ± 0.64 | 59.33 ± 0.37 | 88.40 ± 1.32 | 57.14 ± 1.04 | 57.46 ± 0.55 | 88.63 ± 5.34 | 66.18 ± 0.65 | 63.71 ± 0.96 | 94.32 ± 2.21 |
| | OE | 66.46 ± 0.46 | 61.55 ± 0.35 | 85.22 ± 0.98 | 83.80 ± 8.05 | 77.71 ± 6.62 | 61.70 ± 18.03 | 70.26 ± 6.61 | 66.10 ± 3.81 | 82.98 ± 13.04 |
| | GOLD | 50.36 ± 0.60 | 50.33 ± 0.15 | 95.14 ± 0.59 | 45.88 ± 9.02 | 52.15 ± 1.14 | 96.87 ± 0.54 | 44.74 ± 2.94 | 51.42 ± 1.95 | 96.76 ± 3.39 |
| | GEBM | 59.00 ± 3.76 | 54.60 ± 2.18 | 92.20 ± 0.64 | 42.60 ± 16.22 | 64.90 ± 6.27 | 66.20 ± 13.76 | 51.60 ± 13.96 | 58.80 ± 3.44 | 80.10 ± 7.83 |
| | SISPDE-RBF | 54.00 ± 3.21 | 53.70 ± 2.16 | 92.58 ± 0.95 | 52.97 ± 1.65 | 56.85 ± 1.41 | 95.92 ± 1.19 | 51.56 ± 3.24 | 56.60 ± 2.22 | 96.01 ± 2.21 |
| | **SISPDE** | 64.77 ± 1.60 | 62.08 ± 1.19 | 88.40 ± 4.20 | 97.17 ± 3.29 | 96.63 ± 4.28 | 5.08 ± 13.36 | 81.96 ± 2.58 | 79.51 ± 3.34 | 30.99 ± 5.38 |
| Questions | GCN | 44.43 ± 0.74 | 50.71 ± 0.21 | 94.33 ± 0.59 | 46.49 ± 0.99 | 56.60 ± 0.40 | 100.00 ± 0.00 | 50.56 ± 0.03 | 50.25 ± 0.00 | 94.82 ± 0.03 |
| | Odin | 60.76 ± 0.83 | 58.56 ± 1.48 | 82.88 ± 0.65 | 67.10 ± 8.91 | 69.30 ± 4.26 | 99.52 ± 0.63 | 55.75 ± 0.43 | 60.49 ± 0.41 | 93.55 ± 0.04 |
| | Mahalanobis | 60.71 ± 1.56 | 57.08 ± 1.58 | 93.59 ± 4.64 | 72.42 ± 1.82 | 62.40 ± 2.11 | 78.04 ± 7.56 | 52.23 ± 3.11 | 56.54 ± 1.84 | 91.70 ± 2.23 |
| | GNNSafe | 59.62 ± 0.33 | 56.71 ± 0.46 | 90.45 ± 0.34 | 56.26 ± 0.78 | 67.62 ± 0.07 | 89.46 ± 2.83 | 55.57 ± 1.08 | 56.83 ± 1.58 | 94.42 ± 0.63 |
| | GNSD | 55.50 ± 2.90 | 54.28 ± 0.55 | 94.25 ± 0.26 | 43.85 ± 0.30 | 54.67 ± 0.24 | 99.29 ± 0.13 | 53.79 ± 0.47 | 53.64 ± 0.82 | 93.09 ± 0.40 |
| | MSP | 41.82 ± 1.50 | 50.50 ± 0.50 | 94.77 ± 1.60 | 46.45 ± 0.66 | 57.13 ± 0.32 | 100.00 ± 0.00 | 50.61 ± 0.05 | 50.94 ± 1.19 | 94.84 ± 0.02 |
| | OE | 61.59 ± 0.89 | 57.59 ± 0.32 | 83.28 ± 1.20 | 74.20 ± 9.40 | 75.74 ± 7.14 | 81.98 ± 11.30 | 50.69 ± 0.06 | 50.03 ± 0.02 | 94.71 ± 0.09 |
| | GOLD | 46.73 ± 1.16 | 50.66 ± 0.25 | 94.55 ± 1.30 | 54.15 ± 1.09 | 73.45 ± 1.86 | 99.40 ± 0.05 | 50.07 ± 0.16 | 50.00 ± 0.00 | 97.03 ± 1.09 |
| | GEBM | 51.60 ± 0.15 | 50.40 ± 0.00 | 97.90 ± 0.13 | 0.00 ± 0.00 | 0.00 ± 0.00 | 100.00 ± 0.00 | 67.20 ± 0.72 | 66.70 ± 0.71 | 91.20 ± 0.30 |
| | SISPDE-RBF | 54.30 ± 0.97 | 54.45 ± 1.38 | 94.47 ± 0.43 | 53.91 ± 0.72 | 61.40 ± 0.39 | 99.53 ± 0.02 | 50.38 ± 0.04 | 50.48 ± 0.06 | 94.61 ± 0.14 |
| | **SISPDE** | 61.95 ± 2.45 | 59.68 ± 1.45 | 84.52 ± 1.78 | 74.53 ± 1.33 | 79.16 ± 1.41 | 71.87 ± 1.68 | 56.14 ± 0.39 | 57.42 ± 2.38 | 92.17 ± 1.48 |
| Tolokers | GCN | 48.73 ± 5.26 | 51.59 ± 2.14 | 95.32 ± 1.15 | 75.30 ± 6.32 | 75.98 ± 3.39 | 99.90 ± 0.20 | 55.68 ± 9.83 | 63.33 ± 3.63 | 93.51 ± 3.52 |
| | Odin | 59.09 ± 1.32 | 55.90 ± 0.63 | 86.52 ± 0.70 | 69.84 ± 4.04 | 80.62 ± 1.49 | 100.00 ± 0.00 | 65.70 ± 1.85 | 69.33 ± 1.14 | 94.49 ± 0.35 |
| | Mahalanobis | 55.08 ± 1.85 | 51.23 ± 0.84 | 89.30 ± 1.26 | 86.67 ± 3.40 | 54.29 ± 0.72 | 80.44 ± 7.39 | 53.54 ± 1.43 | 50.85 ± 0.06 | 89.52 ± 1.69 |
| | GNNSafe | 63.92 ± 0.75 | 63.01 ± 0.48 | 89.77 ± 0.54 | 97.56 ± 0.13 | 96.58 ± 0.24 | 1.88 ± 0.20 | 73.43 ± 5.23 | 71.10 ± 3.45 | 90.88 ± 2.20 |
| | GNSD | 51.88 ± 10.56 | 51.64 ± 1.23 | 95.08 ± 0.84 | 66.77 ± 3.18 | 52.67 ± 2.53 | 100.00 ± 0.00 | 42.63 ± 1.07 | 58.32 ± 3.22 | 98.09 ± 0.40 |
| | MSP | 45.54 ± 1.62 | 50.09 ± 0.06 | 96.22 ± 0.61 | 87.46 ± 9.28 | 85.35 ± 8.88 | 76.25 ± 37.60 | 50.70 ± 1.27 | 59.73 ± 0.29 | 95.90 ± 1.20 |
| | OE | 71.02 ± 0.70 | 66.18 ± 0.68 | 83.26 ± 0.85 | 75.64 ± 8.40 | 81.92 ± 6.93 | 65.58 ± 25.71 | 62.49 ± 0.95 | 62.81 ± 2.65 | 92.81 ± 2.65 |
| | GOLD | 49.63 ± 2.17 | 50.90 ± 0.54 | 94.47 ± 1.20 | 61.33 ± 40.91 | 76.17 ± 22.25 | 60.00 ± 48.99 | 47.03 ± 16.96 | 55.02 ± 5.56 | 93.13 ± 1.49 |
| | GEBM | 56.80 ± 0.72 | 53.00 ± 0.32 | 96.60 ± 0.26 | 1.80 ± 0.30 | 90.00 ± 0.17 | 98.20 ± 0.30 | 50.00 ± 1.44 | 50.50 ± 0.16 | 97.50 ± 0.79 |
| | SISPDE-RBF | 60.63 ± 8.90 | 59.26 ± 6.16 | 92.36 ± 7.94 | 93.12 ± 1.23 | 93.99 ± 3.90 | 99.98 ± 0.04 | 76.62 ± 13.62 | 74.52 ± 12.59 | 76.50 ± 33.65 |
| | **SISPDE** | 68.40 ± 1.67 | 64.15 ± 4.59 | 89.07 ± 6.90 | 92.75 ± 8.31 | 89.76 ± 1.43 | 25.89 ± 3.88 | 95.23 ± 8.86 | 91.73 ± 1.26 | 12.53 ± 3.95 |
| Amazon Ratings | GCN | 50.25 ± 0.04 | 52.32 ± 0.01 | 94.87 ± 0.00 | 85.93 ± 1.31 | 77.24 ± 0.30 | 56.88 ± 5.96 | 49.86 ± 0.02 | 50.14 ± 0.02 | 95.20 ± 0.04 |
| | Odin | 51.25 ± 0.42 | 52.49 ± 0.40 | 96.06 ± 0.15 | 35.25 ± 6.61 | 53.81 ± 0.64 | 99.99 ± 0.01 | 51.37 ± 0.62 | 51.28 ± 0.18 | 93.69 ± 0.11 |
| | Mahalanobis | 51.25 ± 0.42 | 51.02 ± 0.12 | 93.31 ± 0.42 | 87.72 ± 0.00 | 63.10 ± 0.00 | 69.31 ± 0.00 | 56.28 ± 1.25 | 53.71 ± 0.54 | 87.31 ± 0.06 |
| | GNNSafe | 51.78 ± 0.34 | 51.58 ± 0.22 | 93.74 ± 0.06 | 33.25 ± 0.13 | 51.66 ± 0.21 | 97.47 ± 0.09 | 50.58 ± 0.30 | 50.73 ± 0.22 | 94.54 ± 0.38 |
| | GNSD | 53.89 ± 0.36 | 53.28 ± 0.55 | 92.58 ± 0.45 | 87.89 ± 0.55 | 86.98 ± 0.31 | 88.15 ± 2.36 | 53.34 ± 0.48 | 52.54 ± 0.42 | 93.56 ± 0.23 |
| | MSP | 50.72 ± 0.15 | 52.14 ± 0.03 | 94.77 ± 0.20 | 86.83 ± 0.66 | 75.60 ± 2.18 | 50.87 ± 3.22 | 50.09 ± 0.03 | 50.27 ± 0.08 | 94.74 ± 0.11 |
| | OE | 52.24 ± 0.21 | 52.09 ± 0.09 | 93.46 ± 0.12 | 88.95 ± 1.91 | 79.51 ± 3.42 | 46.78 ± 5.20 | 50.56 ± 0.12 | 50.46 ± 0.08 | 94.49 ± 0.07 |
| | GOLD | 49.89 ± 0.62 | 50.41 ± 0.32 | 95.25 ± 0.96 | 84.95 ± 8.40 | 83.22 ± 9.45 | 78.79 ± 37.18 | 50.13 ± 0.07 | 50.15 ± 0.03 | 95.05 ± 0.44 |
| | GEBM | 57.20 ± 2.30 | 52.60 ± 1.01 | 85.90 ± 1.66 | 35.90 ± 6.29 | 61.10 ± 3.21 | 73.50 ± 6.64 | 51.20 ± 4.00 | 61.10 ± 2.31 | 74.90 ± 4.37 |
| | SISPDE-RBF | 52.62 ± 0.49 | 52.31 ± 0.46 | 93.51 ± 0.11 | 88.12 ± 0.07 | 87.26 ± 0.36 | 89.53 ± 0.34 | 51.35 ± 0.32 | 51.37 ± 0.31 | 94.01 ± 0.41 |
| | **SISPDE** | 54.19 ± 0.45 | 53.94 ± 0.24 | 91.57 ± 0.50 | 93.73 ± 0.33 | 89.24 ± 1.45 | 45.49 ± 1.88 | 53.36 ± 0.48 | 56.15 ± 1.32 | 93.30 ± 1.76 |

Table 1: Out-of-Distribution detection accuracy (DET-ACC) (↑), AUC (↑), and FPR95 (↓) (best and runner-up) on graph datasets. Our model consistently outperforms baseline models on all datasets and on different OOD types.

| Dataset | Minesweeper | Tolokers | Questions | Roman Empire |
|---|---|---|---|---|
| Result Without Rewiring | 53.97/53.20/92.80 | 48.73/51.59/95.32 | 44.43/**50.71**/**94.33** | 53.15/53.12/92.49 |
| Result With Rewiring | **59.64**/**57.15**/**89.10** | **64.91**/**61.92**/**86.28** | **56.56**/50.24/94.38 | **64.53**/**55.48**/**87.16** |

Table 2: GCN Performance with and without rewiring on low LI datasets (AUC/DET-ACC/FPR95)

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

# A Construction of Spacetime Noise on Graph

In this section, we provide the detailed formulation for defining the $\Phi$-Wiener process in the main paper and its associated stochastic partial differential equation (SPDE) on graph. We introduce the mathematical tools of differential operators on graph, the formulation of a PDE on graph, and the definition of a noise process on graph. For each subsection, we provide the proofs for theorem 1 to 3 in the main paper.

## A.1 Graph Message Passing as PDE

The analogy between Message Passing Neural Networks and Partial Differential Equations has been widely explored, with pioneering work in [10], which systematically defined differential operators on graph. In particular, the graph incidence matrix $G$ is used to model divergence and the gradient operator, while $GG^T = L$, the graph Laplacian matrix, is used to model the Laplace operator. Multiple existing works such as GRAND [5] and GREAD [7] formulate MPNNs as discretization of partial differential equations on the graph domain, with the special formulation of the graph convolution operation as a heat diffusion kernel. This formulation allows them to use an ODE-solver to perform message passing, making the MPNN continuous in depth. In the most basic case, for the node feature $x$ on graph, we have the GRAND-based message passing as:

$$\frac{\partial x(t)}{\partial t} = div[G(x(t), t)\nabla x]$$

which follows the heat equation $\frac{\partial x}{\partial t} = \Delta x$. For graph data, $\nabla x$ is defined as a matrix whose entries are the difference between two node feature: $(\nabla x)_{ij} = x_i - x_j$ and the divergence operator is the aggregation of signals over the neighborhood of a node. In the GRAND formulation, $G$ is a GNN layer while $t$ is the continuous index for layers of the GNN, and the entire message passing that updates node features can be written as an integral equation. For GRAND, the vector field is written as $\partial x/\partial t = (Ax - I)x$, a random walk operation. The full forward pass also incorporates an encoder and decoder network $(\phi, \psi)$ to project the individual node features into a latent space, which results in the following update rule:

$$H^{(0)} = \phi(x)$$
$$H^{(T)} = H^{(0)} + \int_0^T \frac{\partial}{\partial t} H(t)dt = H^{(0)} + \int_0^T (AH(t) - I)H(t)dt$$
$$x^{(T)} = \psi(H^{(T)})$$

This formulation aligns with the neural model known as latent graph ODE, which also uses a GNN as the parametric function for the vector field, while first projecting the input feature into a latent space. The author argued that this formulation can help alleviate over-smoothing and provides a principled way to design GNN message passing schemes, despite the challenge of model training and hyperparameter-tuning.

Beyond the heat equation formulation, various other PDE-based message passing schemes have been proposed, such as reaction-diffusion equation [7], transport equation [39] and the Schrödinger equation [33]. In [26], the author motivated the case of a stochastic heat equation, which appends a $Q$-Wiener process after the heat equation, and was implemented in the form similar to a latent graph Stochastic Differential Equation [3].

## A.2 $Q$-Wiener process and Cylindrical Wiener process

We provide the mathematical definition of two types of space time noise process crucial for the construction of stochastic PDEs.

**Definition 3** (Brownian Motion). *A standard Brownian motion (also called Wiener process) is a stochastic process $\{\beta(t)\}_{t\geq 0}$ defined on a filtered probability space $(\Omega, \mathcal{F}_t, P)$ with the following properties:*

1. *$\beta(0) = 0$.*

2. *$t \mapsto W_t$ is continuous in $t$, almost surely.*

*3. The process has independent increment: $\forall 0 \leq t_1 \leq t_2 \leq \cdots \leq t_n$, the increments $\beta(t_n) - \beta(t_{n-1}), \beta(t_{n-1}) - \beta(t_{n-2}), \cdots, \beta(t_2) - \beta(t_1)$ are independent random variables.*

*4. $\forall s > t \geq 0, \beta(s) - \beta(t) \sim \mathcal{N}(0, t - s)$.*

For the theory of SPDE, the crucial concept is random variable in Hilbert space, which gives rise to the notion of trace-class operators and $Q$-Wiener process.

**Definition 4** (Trace-Class Operators)**.** *Let $H$ be a separable Hilbert space with a complete orthonormal eigenfunction basis $\{\psi_k\}_{k \in \mathbb{N}}$ and $Q : H \to H$ a linear operator, then $Q$ is trace-class if:*

$$Tr(Q) = \sum_{k=1}^{\infty} \langle Q\psi_k, \psi_k \rangle < \infty$$

**Definition 5** (Karhunen-Loève Expansion)**.** *Let $X_t$ be a zero-mean and square-integrable stochastic process defined on $(\Omega, \mathcal{F}, \mathcal{F}_t, \mathbb{P})$ and indexed over a closed and bounded $[a, b]$ with covariance function $K_X(s, t)$, then if $K_X$ is a Mercer kernel, and let $e_k$ be an orthonormal basis of the Hilbert-Schmidt operator on $L^2[a, b]$ with eigenvalues $\lambda_k$, then $X_t$ admits the expansion:*

$$X_t = \sum_{k=1}^{\infty} Z_k e_k(t),$$

*where $\{Z_k\}$ are pairwise orthogonal random variables with zero mean and variance $\lambda_k$, and the series converges to $X_t$ uniformly in mean square.*

**Definition 6** ($Q$-Wiener process)**.** *Let $Q : H \to H$ be a trace-class operator that's symmetric and non-negative, then a $H$-valued stochastic process $\{W(t) : t \geq 0\}$ on the filtered probability space $(\Omega, \mathcal{F}, \mathcal{F}_t, \mathbb{P})$ is a Q-Wiener process if:*

*1. $W(0) = 0$ almost surely.*

*2. $W(t; \omega)$ is a continuous sample trajectory in $H$ for each $\omega \in \Omega$.*

*3. $W(t)$ is $\mathcal{F}_t$-adapted and has independent increments $W(t) - W(s)$ for $s < t$.*

*4. $W(t) - W(s) \sim \mathcal{N}(0, (t - s)Q)$ for all $0 \leq s \leq t$.*

Using the Karhunen-Loève expansion, one can establish that a $W(t)$ is a $Q$-Wiener process if and only if $\forall t \geq 0$:

$$W(t) = \sum_{j=1}^{\infty} \sqrt{\lambda_j} \psi_j \beta_j(t)$$

where $\beta_j(t)$ is i.i.d. Brownian motions, $\psi_i$ are orthonormal eigenfunctions of the Hilbert space, and the series converge in $L^2(\Omega, H)$ [41].

When $Q = I$, then $Q$ is no longer trace class in $H$, so othe series above does not converge in $L^2(\Omega, H)$. In this case we have the Cylindrical Wiener process:

**Definition 7** (Cylindrical Wiener Process)**.** *Let $H$ be a separable Hilbert space. A Cylindrical Wiener process (spacetime white noise) is a $H$-valued stochastic process $\{W(t) : t \geq 0\}$ defined by:*

$$W(t) = \sum_{k=1}^{\infty} \psi_k \beta_k(t)$$

which is simply the case for $Q = I$.

Note that in our construction of the $\Phi$-Wiener process on graph (Definition 2), we have the analogous Hilbert space formulation of:

$$W(t) = \sum_{k=1}^{\infty} \phi(\lambda_j) \psi_j \beta_j(t)$$

and the convergence depends heavily on the behavior of the eigenvalues $\lambda_j$ of the Hilbert space. If the operator $\Phi(\Lambda) = I$, then we recover Cylindrical Brownian motion, and if the infinite series converge in $L^2(H, \Omega)$, we have a $Q$-Wiener process. If it diverges, we have a general class of spacetime white noise.

### A.3 $Q$-Wiener process on Graph

The notion of a $Q$-Wiener process on graph was introduced in [26], where a spectral formulation of the noise is adapted using the spectrum of the graph Laplacian matrix $L = U\Lambda U^T$, where $U = (\mathbf{u}_1, \cdots, \mathbf{u}_{|V|})$ is the orthogonal matrix whose column vectors are the Laplacian eigenvectors and $\Lambda = \text{diag}(\lambda_1, , \cdots, \lambda_{|V|})$ are the Laplacian eigenvalues. Constructing the $Q$-Wiener process follows the classical Karhunen-Loève expansion, which projects standard i.i.d Gaussian noises onto the spectral basis:

$$W(t) = \sum_{k=1}^{|V|} \langle W(t), \mathbf{u}_k \rangle \mathbf{u}_k \;\; = \sum_{k=1}^{|V|} \sqrt{\lambda_k} \mathbf{u}_k \beta_k(t)$$

This construction has the following properties:

1. $W(t) - W(s) \sim \mathcal{N}(0, (t-s)L)$
2. the process $W(t)$ is trace class and a $Q$-Wiener process.
3. $\beta_k(t) - \beta_k(s) = \frac{1}{\sqrt{\lambda_k}} \langle W(t) - W(s), \mathbf{u}_k \rangle$

**Proof of Proposition 1 in the main paper**:

**Proposition 1**: *Let $G = (V, E)$ be a graph, then the Q-Wiener process defined in Equation 4 results in a spatial covariance structure of $Cov(W_i(t), W_j(t)) = L_{ij}t$, where $i, j \in V$, and $\beta_i(t), \beta_j(t)$ are each independent Brownian motion on $i, j$.*

*Proof.*

$$\text{Cov}(W_i(t), W_j(t)) = \mathbb{E}[W_i(t)W_j(t)] - \mathbb{E}[W_i(t)]\mathbb{E}[W_j(t)]$$

$$= \mathbb{E}\left[ \left( \sum_{k=1}^{|V|} \sqrt{\lambda_k} \mathbf{u}_k(i)\beta_k(t) \right) \left( \sum_{k=1}^{|V|} \sqrt{\lambda_k} \mathbf{u}_k(j)\beta_k(t) \right) \right] - 0$$

$$= \mathbb{E}\left[ \sum_{k=1}^{|V|} \lambda_k \beta_k^2(t) \mathbf{u}_k(i)^T \mathbf{u}_k(j) \right] + \mathbb{E}\left[ \sum_{k \neq l}^{|V|} \lambda_k \lambda_l \beta_k(t)\beta_l(t) \mathbf{u}_k(i)\mathbf{u}_l(j) \right]$$

$$= t \sum_{k=1}^{|V|} \lambda_k \mathbf{u}_k(i)\mathbf{u}_k(j) = t \left[ U\Lambda U^T \right]_{ij} = L_{ij}t$$

$\square$

As can be seen, the $Q$-Wiener process defined by [26] can be seen as stochastic process with spatial covariance structure determined by the graph Laplacian.

### A.4 Gaussian Random Field and Graph Gaussian Process

To imbue the noise process with more structure, the usual approach is to examine the covariance of the Gaussian process. A space time noise process in its essence a process with two parameters: $W(x, t)$ where $x$ usually takes the value from a compact set in a space like $\mathbb{R}^d$. The approach that studies space-time white noise from the perspective of compact sets in space and time is pioneered by Whittle [54] and Walsh [51]. In particular, if we freeze time and study the spatial component, we obtain a **Random Field** $W(x)$, where $x$ takes value from a compact set $U \in \mathbb{R}^d$. If the joint distribution of any finite subset of spatial noises admit a multivariate Gaussian, then the random field is known as a **Gaussian Random Field (GRF)**. The study of random field as a solution of SPDE

can be found in [54, 27], where it is known that the GRF with the Matérn Covariance kernel can be obtained by solving the following SPDE:

$$\left(\frac{2\nu}{\kappa^2} - \Delta\right)^{\frac{\nu}{2} + \frac{d}{4}} f = \mathcal{W}$$

where $\mathcal{W}$ is the Gaussian white noise re-normalized by certain constants [4]. On the graph domain, a truncated version of the SPDE above can be formualted by replacing $\mathcal{W} \sim \mathcal{N}(0, \mathbf{I})$ and with the equation:

$$\left(\frac{2\nu}{\kappa^2} - \Delta\right)^{\frac{\nu}{2}} f = \mathcal{W}$$

the Gaussian process as a result of this equation can be represented using the spectral decomposition, similar to the Karhunen-Lèove expansion approach. This was solved in [4] as:

$$f \sim \mathcal{N}\left(0, \left(\frac{2\nu}{\kappa^2} + \Delta\right)^{-\nu}\right)$$

where the covariance kernel is known as the Matérn kernel ($M$), which admits a spectral representation of:

$$M = U\Phi(\Lambda)U^T, \quad \Phi(\lambda) = \left(\frac{2\nu}{\kappa^2} + \lambda\right)^{\nu/2}$$

The parameter $\nu$ controls the smoothenss of the spectrum, with $\nu \to \infty$ corresponding to the heat kernel $\exp(-\frac{\kappa^2}{2}\lambda)$ and small values of $\nu$ resulting in a rough spectrum and the resulting Karhunen-Loève expansion not converging in $L^2(\mathbb{R}^d, \Omega)$.

### A.5  $\Phi$-Wiener process on Graph

Motivated by the limitation of the $Q$-Wiener process on graph and the covariance structure based approach in Gaussian Random Field, we proposed the $\Phi$-Wiener process, which performs the spectrum transformation on the eigenvalues induced by a predefined covariance kernel function.

**Definition($\Phi$-Wiener process on graph).** Let $G = (V, E)$ be an undirected graph and $\Delta$ its graph Laplacian. Let $\{\mathbf{u}_k\}_{i=1}^{|V|}, \{\lambda_k\}_{k=1}^{|V|}$ be its eigenvectors and eigenvalues, respectively, and let $\phi : \mathbb{R} \to \mathbb{R}$ be a scalar function with $\Phi$ its matrix functional form, then a $\Phi-$Wiener process is defined by the (truncated) Karhunen-Loève expansion:

$$W(t) = \sum_{k=1}^{|V|} \phi(\lambda_k)\mathbf{u}_k\beta_k(t) \tag{15}$$

In the experimental part of the paper, we in particular studied the case when $\phi$ corresponds to the Matérn kernel, while the construction can be applied to any positive semidefinite kernel matrix $K$ (or the kernel function in the Hilbet space). Here we present the proof for Theorem 2 in the main paper:

**Proof for Theorem 1 in the main paper**:

**Theorem 1**: *Let $G = (V, E)$, and $\Delta$ its graph Laplacian with spectral decomposition $\Delta = U\Lambda U^T$, then for the construction in definition 2, if the matrix $K = U\Phi(\Lambda)U^T$ is positive definite, then $W(t) - W(s) \sim \mathcal{N}(\mathbf{0}, (t - s)K)$ for $t > s$ and the spatial-temporal covariance of noise process on two nodes $j, k$ can be written as $Cov(W_i(t), W_j(s)) = (t \wedge s)K_{ij}$, where $(t \wedge s) = min(t, s)$.*

*Proof.* The result that $W(t) - W(s) \sim \mathcal{N}(\mathbf{0}, (t - s)K)$ follows since the truncated KL expansion is equivalently embedding on the matrix $K = U\Phi(\Lambda)U^T$, and the form follows from the definition of $Q$-Wiener process in Appendix A.3. As for the covariance structure, we can obtain the result by

adjusting the proof for proposition 1:

$$\mathrm{Cov}(W_i(t), W_j(s)) = \mathbb{E}[W_i(t)W_j(s)] - \mathbb{E}[W_i(t)]\mathbb{E}[W_j(s)]$$

$$= \mathbb{E}\left[\left(\sum_{k=1}^{|V|} \phi(\lambda_k)\mathbf{u}_k(i)\beta_k(t)\right)\left(\sum_{k=1}^{|V|} \phi(\lambda_k)\mathbf{u}_k(j)\beta_k(s)\right)\right] - 0$$

$$= \mathbb{E}\left[\sum_{k=1}^{|V|} \phi^2(\lambda_k)\beta_k(t)\beta_k(s)\mathbf{u}_k(i)^T\mathbf{u}_k(j)\right] + \mathbb{E}\left[\sum_{k\neq l}^{|V|} \phi(\lambda_k)\phi(\lambda_l)\beta_k(t)\beta_l(s)\mathbf{u}_k(i)\mathbf{u}_l(j)\right]$$

$$= \mathbb{E}[\beta_k(t)\beta_k(s)]\sum_{k=1}^{|V|} \phi^2(\lambda_k)\mathbf{u}_k(i)\mathbf{u}_k(j) = \mathbb{E}[\beta_k(t)\beta_k(s)]\left[U\Phi(\Lambda)U^T\right]_{ij} = (t \wedge s)K_{ij}.$$

$\square$

Notice that with the specific choice of $\phi$ in our paper, the covariance structure on the spatial domain should correspond to the Matérn kernel.

As a theoretical interest, we can analogously look at the general Hilbert space construction of the $\Phi$-Wiener process, by looking at the full KL expansion:

$$W(t) = \sum_{k=1}^{\infty} \phi(\lambda_k)\psi_k\beta_k(t)$$

where $\{\lambda_k, \psi_k\}_{\mathbb{N}}$ are now the orthonormal eigenbasis of a separable Hilbert space $H$. We stated the property of this stochastic process in Theorem 2 of the main paper. Here we restate the theorem and provide a proof.

**Proof of Theorem 2 in the main paper**:

**Theorem 2**: *Let $\mathcal{H}$ be a separable Hilbert space and $(\Omega, \mathcal{F}, \mathcal{F}_t, \mathbb{P})$ a filtered probability space. Let $Q'$ be a trace-class operator in $\mathcal{H}$ and $\Phi$ a function operator that scalar-transforms the eigenvalues $\{\lambda_i\}$ of $Q'$, then if $\sum_{i=1}^{\infty} \sqrt{\phi(\lambda_i)} < \infty$, the induced $\Phi$-Wiener process is a $Q$-Wiener process with the trace class operator $Q = \Phi(Q')$. Moreover, the transformation for Matérn kernel induces a $Q$-Wiener process when $\nu > d$ for a random field taking value in Compact $U \subset \mathbb{R}^d$.*

*Proof.* The first part of the theorem simply states that if the operator induced by $\Phi(Q')$ is trace class, then the resulting Wiener process is a $Q$-Wiener process with $Q = \Phi(Q')$. This is true by definition. The main proof concerns the case for the Matérn kernel, which we aim to prove here.

Let $\Delta$ be the Laplace-Beltrami operator, which is the Hilbert space analogy of the Laplacian matrix, then we need to show that the Matérn kernel's induced spectral decomposition results in a trace-class operator. This is equivalent to proving that the following infinite series converge:

$$\sum_{n=1}^{\infty} \left(\frac{2\nu}{\kappa^2} + \lambda_n\right)^{-\nu-\frac{d}{2}} < \infty$$

Where the additional term with $d$ comes from the Whittle SPDE for $\mathbb{R}^d$ [4]:

$$\left(\frac{2\nu}{\kappa^2} + \Delta\right)^{\frac{\nu}{2}+\frac{d}{4}} f = \mathcal{W}$$

which ensures regularity and existence of solution. Proving the convergence of the series requires analyzing the asymptotic behavior of the Laplacian eigenvalues $\{\lambda_n\}$. Assume that the boundary condition is Neumann, then the Laplace problem can be written in spectral form as:

$$-\Delta u_k = \lambda_k u_k, \quad u_k \mid_{\partial U} = 0$$

with the eigenvalues satisfying:

$$0 \leq \lambda_1 \leq \lambda_2 \leq \cdots \leq \cdots < \infty$$

Let the eigenvalue counting function $N(\lambda)$ be:

$$N(\lambda) = \#\{k \in \mathbb{N} : \lambda_k \leq \lambda\}$$

then $N(\lambda_n) = n$. Now the classical Weyl's law [19] says that:

$$N(\lambda) \sim c \cdot vol(U)\lambda^{d/2} \sim O(\lambda^{d/2})$$

where $vol(U) < \infty$ is the Lebesgue measure of $U$, which is finite since $U$ is compact. We hence arrive at the following relationship:

$$n \sim O(\lambda_n^{d/2}) \implies \lambda_n \sim O(n^{2/d})$$

Now with the asymptotic behavior of $\lambda_k$ clear, we refer back to the original infinite series, whose convergence is controlled by:

$$\sum_{n=1}^{\infty} (n^{2/d})^{-\nu - \frac{d}{2}} = \sum_{n=1}^{\infty} n^{-(\frac{2\nu}{d}+1)}$$

which converges if and only if:

$$-\frac{2\nu}{d} + 1 < -1 \implies \nu > d$$

$\square$

# B  Stochastic PDE with $\Phi$-Wiener process on Graph

In this section we analyze the conditions for a SPDE to have a unique mild solution. In particular, we analyze the case when the SPDE is driven by the $\Phi$-Wiener process defined in the section above, and provide a proof for theorem 4 in the main paper. In the end, we also provide the derivation of the graph ODE implementation of the SPDE using the Wong-Zakai approximation theorem.

## B.1  Existence and Uniqueness of Solution to SPDE

In this section we provide the technical details about the existence and uniqueness of the solution to a stochastic PDE system driven by a $Q$-Wiener process and then justify the SPDE system driven by a $\Phi$-Wiener process in our case by proving Theorem 3 in the main paper:

**Theorem 3**: *Let $\mathcal{H}$ be a separable Hilbert space, and $(\Omega, \mathcal{F}, \mathcal{F}_t, \mathbb{P})$ be a filtered probability space and $W(x, t)$ be a $\mathcal{F}_t$-adapted space-time stochastic process whose spatial covariance kernel is given by the Matérn kernel on a closed and bounded domain $D \subset \mathbb{R}^d$, then if $v > d$ and $\mathbf{H}(0)$ is square-integrable and $\mathcal{F}_0$-measurable, the SPDE of the form:*

$$d\mathbf{H}(t) = \Big(\mathcal{L}\mathbf{H}(t) + F(\mathbf{H}(t))\Big)dt + G(\mathbf{H(t)})dW(t)$$

*admits a unique mild solution:*

$$\mathbf{H}(t) = \exp(t\mathcal{L})\mathbf{H}(0) + \int_0^t \exp((t-s)\mathcal{L})F(\mathbf{H}(s))ds + \int_0^t \exp((t-s)\mathcal{L})G(\mathbf{H}(s))dW(s)$$

*Where $\mathcal{L}$ is a bounded linear operator generating a semigroup $\exp(t\mathcal{L})$ and $F, G$ are global Lipschitz functions.*

*Proof.* The condition that $\nu > d$ ensures that the Space time noise with Matérn kernel in our case is a $Q$-Wiener measure (from Theorem 2). This is applicable since a closed and bounded domain $D \subset \mathbb{R}^d$ is a compact set. The theorem then becomes the classical result of existence and uniqueness of mild solution of semilinear SPDE driven by $Q$-Wiener process. The technical details of the proof can be found in [16] Theorem 6.4. Since the whole proof requires a series of technical build ups, we refer the readers to the monograph by Hairer [16]. $\square$

## B.2 Chebyshev Approximation of Matérn Kernel

For the practical implementation in section 3.3, we propose to use the Chebyshev polynomial to approximate the matrix function $\Phi(\Delta)$ when eigendecomposition becomes too expensive for the graph Laplacian $\Delta$. Here we provide the theoretical justification for the approximation bound derived in Theorem 4.

**Theorem 4**: *Let $G = (V, E)$ be a graph, and let the Matérn kernel on graph be defined in equation 5, then the Chebyshev approximation in equation 13 results in an error bound of $\mathcal{O}\left(\left(\frac{\kappa^2 d_{max}}{16\nu + \kappa^2 d_{max}}\right)^m\right)$, where $m$ is the degree of the Chebyshev polynomial and $d_{max}$ is the maximum degree of the graph.*

*Proof.* Equation 5 indicates that the function to approximate is:

$$f(\lambda) = \left(\frac{2\nu}{\kappa^2} + \lambda\right)^{-\nu}$$

which is generally analytical except at the singularity of $\lambda = -\frac{2\nu}{\kappa^2}$. However, since the eigenvalues are non-negative for the graph Laplacian and $\nu, \kappa \geq 0$, the function is analytical in the entire domain of $\lambda$. In practice, however, we scale $\lambda$ to the range $[-1, 1]$ for the graph Laplacian before the Chebyshev approximation. Hence without loss of generality, suppose

$$\tilde{\Delta} = \frac{2\Delta - (\lambda_{max} - \lambda_{min})I}{\lambda_{max} - \lambda_{min}}$$

Then analysis concerns $f(\lambda)$ on the interval $[-1, 1]$. Since $f(\lambda)$ is analytic on this interval, we can use result in [9, 46]: Let $E_\rho$ be the Bernstein Ellipsis over which $f(\lambda)$ is analytic. Let $\rho > 1$, then a Chebyshev polynomial of degree $K$ has approximation error of:

$$|g(x) - g_K(x)| \leq \frac{CM(\rho)}{\rho - \rho^{-1}}\rho^{-K} \leq C\frac{M(\rho)}{\rho - 1}\rho^{-K}$$

where $M(\rho) = \max_{\lambda \in E_\rho}|f(\lambda)|$ is the bound for $f(\lambda)$ over the Bernstein Ellipsis, defined in the complex plane as:

$$E_\rho = \{z \in \mathbb{C} : z = \frac{1}{2}(\rho e^{i\theta} + \rho^{-1}e^{-i\theta}), \theta \in (0, 2\pi)\}$$

The singularity of the function $f(\lambda)$ is at $\lambda = -\frac{2\nu}{\kappa^2}$, which should lie outside of the rescaled eigenvalue range $[-1, 1]$. Let $x_s$ be the point of singularity, then:

$$x_s = \frac{2\lambda - (\lambda_{max} + \lambda_{min})}{\lambda_{max} - \lambda_{min}} = \frac{2(-\frac{2\nu}{\kappa^2}) - \lambda_{max}}{\lambda_{max}} = -\frac{4\nu}{\kappa^2\lambda_{max}} - 1$$

For $\{\lambda_n\}$ the eigenvalues of the graph Laplacian, following the conclusion in [56], we have that $\lambda_{max} \leq \frac{d_{max}}{2}$, where $d_{max}$ is the maximum degree of the graph. Let $\alpha = \frac{8\nu}{\kappa^2 d_{max}}$, then:

$$x_s \leq -\frac{8\nu}{\kappa^2 d_{max}} - 1 := -\alpha - 1$$

and $\rho$ being the distance from the spectrum of $\Delta$ to the nearest singularity, is computed as:

$$\rho = |x_s| + \sqrt{x_s^2 - 1} \geq \alpha + 1 + \sqrt{(\alpha + 1)^2 - 1} \geq 2\alpha + 1$$

hence we have:

$$\rho \geq 2\alpha + 1 \geq \frac{16\nu}{\kappa^2 d_{max}} + 1$$

In the end, we use the approximation bound in [9] again:

$$|g(x) - g_K(x)| = \mathcal{O}(\rho^{-K}) = \mathcal{O}\left(\left(\frac{1}{\frac{16\nu}{\kappa^2 d_{max}} + 1}\right)^K\right) = \mathcal{O}\left(\left(\frac{\kappa^2 d_{max}}{16\nu + \kappa^2 d_{max}}\right)^K\right)$$

Note that tighter bounds are possible, depending on the bound of choice for $\lambda_{max}$ and $M(\rho)$, but we leave that for future discussion. This bound nevertheless provides the insight that if larger values of $\nu$ results in faster convergence, which naturally follows due to a smoother spectrum of the Laplacian. $\square$

**B.3 Discussion on Node Variance**

Throughout the paper we have focused on the correlation of uncertainty between different nodes due to graph topology and label distribution. Now we briefly discuss on the behavior of node-level uncertainty, characterized by variance of the accumulated noise. For the Matérn kernel $K = U(\frac{2\nu}{\kappa}I + \Lambda)^{-\nu}U^T$, the variance of each node can be computed as:

$$K_{ii} = \sum_{j=1}^{n}(\frac{2\nu}{\kappa^2} + \lambda_j)^{-\nu}U_{ij}^2 \tag{16}$$

This variance depends on the graph topology as the above equation is a function of Laplacian eigenvalues and eigen-vectors, and is not directly clear whether they are small or large by just looking at $\nu$. However, what we can tell is that, fixing all the other terms constant and set $2\nu/\kappa^2 = 1$, then since graph Laplacian eigenvalues are non-negative, the larger the value of $\nu$, the more it seems to penalize the variance term (hence the terminology of a smooth kernel). In addition, if $\nu$ is large, the distribution of variance is more even for neighbors or connected components (if one node has large variance, neighbors also do), whereas if $\nu$ is small, we can have two distant nodes sensitive to the variance change of each other. Ultimately the kernel function controls covariance while variance has more to do with graph topology itself, due to the direct link with the spectrum $\Lambda, U$.

Hence, in our model, the more direct answer to how variance is controlled for each node individually is the integration time in the graph Neural ODE in section 3.3, since longer integration time leads to accumulation of more variance, hence more uncertainty. Analyzing variance is therefore an interesting direction to explore, since we know that more layers of GNN (corresponding to more integration steps) usually lead to over-smoothing of graph signals, but the noise accumulation can seem to introduce variance. Since these two dynamics can interact with each other in intricate ways, it may even be possible that longer integration time doesn't necessarily lead to exploding variance. In our empirical studies, we concluded that learning converges with loss function converging, but it would indeed be interesting to study how the variance of each individual nodes (or sub-regions, since the graphs are usually very large) evolve. We believe that this topic itself can be an interest of extensive future studies.

## C  Experiment details

In this section we provide additional details on the configurations for experiments, including dataset description, baseline description, evaluation metrics, hyperparameter selection ranges, and additional empirical results.

### C.1  Experiment Environment

The experiments are conducted on the Mosaic ML platform with 8 H100 GPUs, and the cluster docker image is the latest AWS image with Ubuntu 20.04. We implement our pipeline using Python 3.12 and PyTorch version 2.5.1 with CUDA 12.4 support.

### C.2  Dataset Description

- **Cora, Pubmed, Coauthor-CS** are three citation graph datasets. Each graph is directed, and nodes are documents, edges are citations. The features are bag-of-words representations. In particular, Cora contains $2,708$ nodes, $5,429$ edges, $1,433$ features and $7$ classes. Pubmed has $19,717$ nodes, $88,648$ edges, $500$ features and $3$ classes. Coauthor-CS contains $18,333$ nodes, $163,788$ edges, $6,805$ features, and $15$ labels. The train-validation-test splits of these three datasets follow the standard practice of previous works. These datasets are considered to be of *Moderate to high label informativeness*, with the particular LI values found in table 3, which is originally in [35]. Coauthor-CS has edge LI to be $0.65$, Cora has edge LI to be $0.59$, and Pubmed has edge LI to be $0.41$.

- **Roman-Empire, Tolokers, Minesweeper, Questions, Amazon-Ratings** are so-called heterophilic datasets [36] that also have low label informativeness, which poses challenge for traditional GNNs. In particular, roman-empire has $22,662$ nodes, $32,927$ edges, $300$

Table 3: Dataset characteristics, more homophilous datasets are above the line. In the paper we focus on $LI_{edge}$, representing edge label informativeness. The table is obtained from [35]

.

| Dataset | $n$ | $|E|$ | $C$ | $h_{edge}$ | $h_{node}$ | $h_{class}$ | $h_{adj}$ | $LI_{edge}$ | $LI_{node}$ |
|---|---|---|---|---|---|---|---|---|---|
| cora | 2708 | 5278 | 7 | 0.81 | 0.83 | 0.77 | 0.77 | 0.59 | 0.61 |
| citeseer | 3327 | 4552 | 6 | 0.74 | 0.72 | 0.63 | 0.67 | 0.45 | 0.45 |
| pubmed | 19717 | 44324 | 3 | 0.80 | 0.79 | 0.66 | 0.69 | 0.41 | 0.40 |
| coauthor-cs | 18333 | 81894 | 15 | 0.81 | 0.83 | 0.75 | 0.78 | 0.65 | 0.68 |
| coauthor-physics | 34493 | 247962 | 5 | 0.93 | 0.92 | 0.85 | 0.87 | 0.72 | 0.76 |
| amazon-computers | 13752 | 245861 | 10 | 0.78 | 0.80 | 0.70 | 0.68 | 0.53 | 0.62 |
| amazon-photo | 7650 | 119081 | 8 | 0.83 | 0.85 | 0.77 | 0.79 | 0.67 | 0.72 |
| lastfm-asia | 7624 | 27806 | 18 | 0.87 | 0.83 | 0.77 | 0.86 | 0.74 | 0.68 |
| facebook | 22470 | 170823 | 4 | 0.89 | 0.88 | 0.82 | 0.82 | 0.62 | 0.74 |
| github | 37700 | 289003 | 2 | 0.85 | 0.80 | 0.38 | 0.38 | 0.13 | 0.15 |
| twitter-hate | 2700 | 11934 | 2 | 0.78 | 0.67 | 0.50 | 0.55 | 0.23 | 0.51 |
| ogbn-arxiv | 169343 | 1157799 | 40 | 0.65 | 0.64 | 0.42 | 0.59 | 0.45 | 0.53 |
| ogbn-products | 2449029 | 61859012 | 47 | 0.81 | 0.83 | 0.46 | 0.79 | 0.68 | 0.72 |
| actor | 7600 | 26659 | 5 | 0.22 | 0.22 | 0.01 | 0.00 | 0.00 | 0.00 |
| flickr | 89250 | 449878 | 7 | 0.32 | 0.32 | 0.07 | 0.09 | 0.01 | 0.01 |
| deezer-europe | 28281 | 92752 | 2 | 0.53 | 0.53 | 0.03 | 0.03 | 0.00 | 0.00 |
| twitch-de | 9498 | 153138 | 2 | 0.63 | 0.60 | 0.14 | 0.14 | 0.02 | 0.03 |
| twitch-pt | 1912 | 31299 | 2 | 0.57 | 0.59 | 0.12 | 0.11 | 0.01 | 0.02 |
| twitch-gamers | 168114 | 6797557 | 2 | 0.55 | 0.56 | 0.09 | 0.09 | 0.01 | 0.02 |
| genius | 421961 | 922868 | 2 | 0.59 | 0.51 | 0.02 | -0.05 | 0.00 | 0.17 |
| arxiv-year | 169343 | 1157799 | 5 | 0.22 | 0.29 | 0.07 | 0.01 | 0.04 | 0.12 |
| snap-patents | 2923922 | 13972547 | 5 | 0.22 | 0.21 | 0.04 | 0.00 | 0.02 | 0.00 |
| wiki | 1770981 | 242605360 | 5 | 0.38 | 0.28 | 0.17 | 0.15 | 0.06 | 0.04 |
| roman-empire | 22662 | 32927 | 18 | 0.05 | 0.05 | 0.02 | -0.05 | 0.11 | 0.11 |
| amazon-ratings | 24492 | 93050 | 5 | 0.38 | 0.38 | 0.13 | 0.14 | 0.04 | 0.04 |
| minesweeper | 10000 | 39402 | 2 | 0.68 | 0.68 | 0.01 | 0.01 | 0.00 | 0.00 |
| workers | 11758 | 519000 | 2 | 0.59 | 0.63 | 0.18 | 0.09 | 0.01 | 0.02 |
| questions | 48921 | 153540 | 2 | 0.84 | 0.90 | 0.08 | 0.02 | 0.00 | 0.01 |

features, and 10 classes. Amazon-ratings has $24,492$ nodes, $93,050$ edges, 300 features, and 5 classes. Minesweeper has $10,000$ nodes, $39,402$ edges, 7 features, and 2 classes. Tolokers has $11,758$ nodes, $519,000$ edges, 10 features, and 2 classes. Questions has $48,921$ nodes, $153,530$ edges, 301 features, and 2 classes. These datasets' LI are in table 3, with roman-empire's LI to be $0.11$, amazon-ratings LI to be $0.04$, minesweeper $0.00$, and questions $0.00$.

Overall we can observe a significant gap of label informativeness between the homophillic datasets and the heterophilic datasets.

## C.3 Label Leave out Details

: For the experiment of label leave out, we follow the common practices for the homophilous datasets; for heterophilous datasets, we either choose the last label or more labels to alleviate class imbalance. Below we present the OOD label choice for all datasets:

- **Cora**: class labels $4, 5, 6$ as IND while class labels $0, 1, 2, 3$ as OOD samples.
- **Pubmed**: class label $1, 2$ as IND samples while class labels $0$ as OOD samples.
- **Citeseer**: class labels $3, 4, 5$ as IND while class labels $0, 1, 2$ as OOD samples.
- **Tolokers**: class $0$ as IND while class $1$ as OOD samples.
- **Roman Empire**: class labels 0-8 as IND samples while class labels 9-17 as OOD samples.
- **Amazon Ratings**: class labels $0, 1, 2$ as IND samples while class labels $3, 4, 5$ as OOD samples.
- **Minesweeper**: class $0$ as IND while class $1$ as OOD samples.
- **Questions**: class $0$ as IND while class $1$ as OOD samples.

## C.4 Baseline Description

- **GCN** [21] is a fundamental GNN architecture that performs spatial graph convolution.
- **ODIN** [25] uses a temperature scaled softmax activation and uses small perturbations to inputs to augment the gap between in-distribution and out-of-distribution data.
- **MSP** [17] uses the density derived from the softmax function to perform OOD detection, with the OOD data to have overall lower density values.
- **Mahalanobis**[23] uses a confidence score based on Mahalanobis distance. They use this score to obtain the class conditional Gaussian distributions with respect to (low- and upper-level) features and to determine IND vs OOD samples.
- **OE** [18] proposes to improve deep anomaly detection by training an anomaly detectors against an auxiliary dataset of outliers and to use the detector later on to detect OOD samples.
- **GNNSafe** [55] proposes to treat the classifier-induced logit as score for an energy based model and to use a label propagation scheme to propagate energy, such that the OOD samples get higher energy and IND samples get lower energy. The model was proposed with or without OOD sample exposure, and we use it for the later case for a fair comparison.
- **GNSD**[26] is an earlier stochastic message passing framework that motivates a stochastic diffusion equation formulation of GNN message passing. It assumes a spatially independent noise and uses aleatoric uncertainty of each node as score for OOD detection.

## C.5 Evaluation Metrics

The evaluation metrics we choose were also used by previous works [55, 26, 22]:

- **DET-ACC**: OOD detection accuracy is the ratio between the test samples that are correctly detected as OOD samples and all test samples. Higher DET-ACC indicates better performance.
- **FPR95** is False Positive Rate at $95\%$ true positive rate. It is the probability that an OOD sample is misclassified when the true positive rate is $95\%$. Smaller FPR95 indicates better performance.
- **AUC** is the Area Under the Curve for the ROC curve. With varying thresholds, it is able to evaluate how the model is able to distinguish between OOD and IND samples. Higher AUC indicates better performance.

## C.6 Hyperparameter Search

For hyperparameter search, we perform grid search on the following parameters:

- kernel smoothness $\nu$: $\{0.1, 0.3, 0.5, 1.0, 3.0, 4.0, 5.0, 10.0, 20.0, 50.0\}$.
- latent dimension: $\{64, 128, 256, 512, 1024\}$.
- learning rate: $\{1e^{-4}, 1e^{-3}, 1e^{-2}\}$
- weight decay:$\{1e^{-4}, 1e^{-3}, 1e^{-2}\}$
- dropout: $\{0.0, 0.5\}$
- Chebyshev polynomial order: $\{30, 40, 50, 60, 80, 100\}$
- training sample times: $\{1, 3, 5, 7\}$

## C.7 Runtime Analysis

In this part we provide the result of running a single forward pass for the models on each datasets. As can be seen from Figure 4, except for two datasets (Pubmed, Questions), our SISPDE's runtime is comparable to other models. We suspect that the reason lies in the matrix multiplication when applying the Cholesky factor / Matérn kernel to the Gaussian noise and the process can be made more efficient by exploring sparse matrix methods, as was also pointed out in [4]. This suggests that our code can be further optimized for more efficient matrix multiplication.

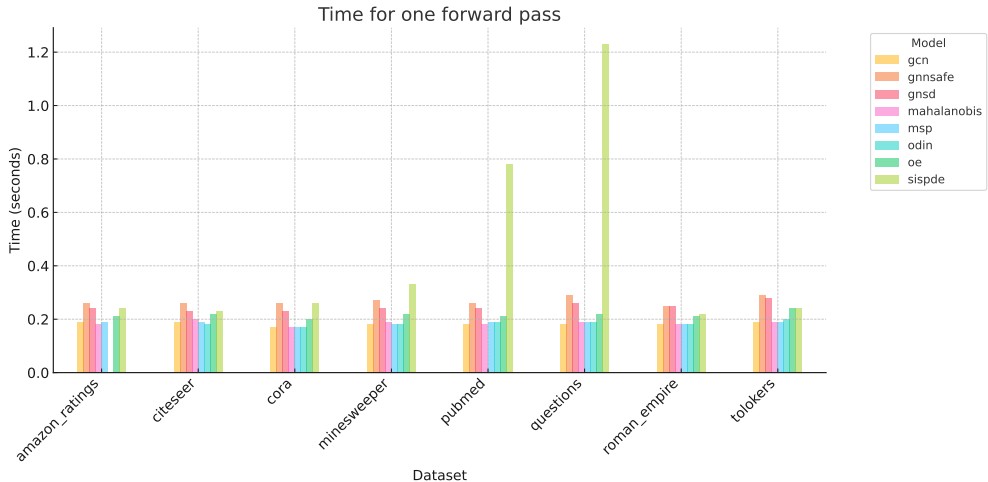

Figure 4: Runtime for one forward pass in seconds for all the models across all the datasets.

## D  Details on Related Works

In the paper we went over related works in the introduction and the background parts. However, due to the page limit, we choose to put detailed related works in the appendix section.

**Uncertainty Estimation on Graph**. Uncertainty Estimation is a booming subject in machine learning, and in the context of graph learning, the community has so far extensively focused on node-level tasks, such as node classification, in which case Out-of-Distribution (OOD) Detection has been one of the main downstream evaluation tasks. For this task, earlier works focus on adding simulated OOD samples, such as OE and MSP[18, 17], or use distribution density based estimates such as Mahalanobis distance [23] and perturbation based methods that aim to modify the distribution of the softmax activation, such as ODIN [25]. Later on, deep generative models were used to directly model the predictive uncertainty of the classification model. Energy based models [55, 29] has been widely adapted, with GNNSafe [55] to explicit model energy propagation on graph to separate the IND samples and OOD samples when these labels are available. However, when the labels are not available during training, models that rely on simulating stochastic processes on graph, such as GNSD [26], usually perform better. Our SISPDE can be seen as an extension and generalization of the GNSD framework, in that we better incorporated the distribution of labels without incorporating explicit label information and our model is mathematically more expressive.

**Physics-Inspired Message Passing Network**. The first paper that draws an analogy between a Message Passing Neural Network (MPNN) and Partial Differential Equation (PDE) is PDE-GCN [10], which systematically defined differential operators on graph. The continuous integral equation formulation of message passing was later expanded by GRAND [5], which motivates message passing as a diffusion process and therefore discretizes the heat equation. Later works such as ACMP[53], GREAD [7], and GRAND++ [48] further expanded the paradigm by introducing more powerful class of PDEs such as classes of reaction-diffusion equations, which explored the message passing mechanism using special properties of these PDE systems. Most recently there has been growing interests to use tools from stochastic analysis, Stochastic Partial Differential Equations, to further expand the class of PDEs. GNSD [26] motivates to use the eigenspace of the graph Laplacian to define a Wiener process on graph, following the approach of SPDE in [38]. Our SISPDE also explores the adaptation of SPDE to message passing. However, we follow more closely to the random field formulation of SPDE in [54] and propose a richer family of space time noise process to drive the message passing process.

**Label Informativeness and Heterophily**. For the task of node classification, there has been a long line of research on the dichotomy of homophilous and heterophilous graphs [31, 32, 35, 36], where it has been observed early that traditional GNNs usually perform poorly on heterophilous graphs. However, it was later discovered that for the datasets proposed to be used for the evaluation task, GNN after sufficient hyperparameter search yields results as competitive as heterophily-specific graph

models, resulting in some questions on the evaluation benchmarks [36]. This discovery motivates the concept of Label Informativeness (LI), which provides a label distribution-based view on the phenomenon of heterophily, with strong correlation between GNN's performance on the dataset and the magnitude of LI [31, 36]. Instead of relying on the homophily vs. heterophily dichotomy, it is therefore more reliable to divide the graph datasets according to label informativeness [35].

**Gaussian Process on Graph**. Gaussian Process models [40] has been wide applications in various fields of machine learning. Although traditionally operating only on Euclidean space, it has been proposed early in [28, 27] that it can be extended to different geometric domains by treating it as solution to certain types of SPDE [54]. On geometric domains, these processes are also known as Gaussian Random Fields [54, 27], and on graph, it was proposed by [4] that Gaussian process can be defined using the graph Laplacian as the discrete representation of the Laplace operator, drawing also connection between the PDE perspective on message passing. The specific kernel in this case is the Matérn Kernel, which allows control of smoothness of the covariance structure. Alternative kernel choices are the symmetric normalized graph Laplacian [4]. In this work we focus on the Matérn kernel since we need a mechanism to explicitly control for kernel smoothness.

