# OpenReview forum: "Uncertainty Estimation on Graphs with Structure Informed Stochastic Partial Differential Equations"
_NeurIPS.cc/2025/Conference — NeurIPS 2025 poster_

### Official Review · Reviewer_iXcv · 2025-06-02

**Clarity:** 2
**Significance:** 3
**Originality:** 2
**Rating:** 5
**Confidence:** 3

**Summary:**

The paper uses a Graph Stochastic Partial Differential Equation to model uncertainty for node classification. The model can represent uncertainty through the stochasticity of a Wiener Process component guided by a Matern Graph Gaussian Process. It consistently outperforms previous models both on homophilic and heterophilic graphs in terms of out-of-distribution detection.

**Questions:**

1. Is $\nu$ a learnable parameter? If not, how is it tuned?
2. Why is a $\nu$ needed to capture "long-range" interactions if the pointwise transformation of the graph's spectrum is already enabled through $\Psi$?
3. Is the term "long-range" interaction justified? From my understanding, what happens is that the kernel promotes high frequency signals (in terms of the graph spectrum), which need not be long-range per se. Would "high-frequency" patterns be a more suitable term?
4. For the leave-out-class setting, the authors also consider some datasets with only two classes. Did the authors train the model only in one class in this setting? I find it very surprising that the model learns anything sensible in this case, as just predicting this one class for any input should minimize the training loss.
5. How is the LI computed? Eq. 1 suggests that one needs access to $p(y_i | y_j)$, but this distribution is unknown as the data-generating process is unknown. Is this distribution estimated as MLE from the entire graph?

**Ethical Concerns:**

["NO or VERY MINOR ethics concerns only"]

**Final Justification:**

The authors resolved all minor concerns and answered questions. Since I still maintain some reservations regarding novelty and clarity of the paper, I raise my score by only one point.

**Limitations:**

yes

**Quality:**

3

**Strengths And Weaknesses:**

## Strenghts
1. The approach is clearly motivated as the GSDN [1] model it builds upon assumes homophily also in the uncertainty by using Laplacian-based propagation to guide the uncertainty. Since heterophilic graphs violate this assumption, a learnable diffusion process is needed.
2. The empirical evaluation of the paper is thorough and methodically sound and shows a clear and consistent improvement over previous work.
3. The paper provides an insightful intuition for the learned covariance that can be interpreted as similar to rewiring the graph to connect nodes with high mutual information in their label distribution.

## Weaknesses
### Major
1. While well-motivated, the approach in this paper is very similar to GSDN. As far as I understand, it only extends on GSDN by introducing pointwise learnable transformations of the eigenvalues of the Graph Laplacian.
2. The paper is sometimes a bit difficult to follow, especially if the reader is not super familiar with Partial Stochastic Differential Equations. My understanding of the paper's flow is: 1) It motivates a spatial kernel different from the Laplacian due to possible high frequencies in the epistemic uncertainty. 2) It introduces pointwise transformations to the graph spectrum and shows that the resulting process is a Wiener process. 2b) This also holds when applying a Matern kernel using the transformed Laplacian. 3) The proposed model is the solution to the resulting GSPDE, realized through forcing.
- I found it difficult to follow the red thread that clearly motivates each of these components. For example, in Theorem 3, where exactly does the pointwise transformation of $\Delta$'s spectrum manifest? What is different from a version where no Matern Kernel (with transformed spectrum?) is used?
- Later on (Eq. 11-13), it seems to me that there is no pointwise transformation of the graph's spectrum anymore and only the Matern kernel (i.e. where is $\Phi$?). Can the authors clarify that?
- Figure 1 shows the Matern kernel (which is used in this work but not a novel contribution) on a synthetic heterophilic graph. What is the takeaway from this figure in terms of the proposed model? Is this to motivate a different diffusion kernel than just the standard Laplacian? If so, how does this motivate the learnable transformation of the graph's spectrum?

### Minor
1. There are some grammatical errors and inconsistencies in the paper (e.g. whether references to equations, figures, etc. are capitalized). I recommend using a grammar-checking tool to fix those.
2. The most recent baseline for UQ in node classification, GEBM [2], is cited but not compared against. Can the authors extend their evaluation to this baseline?
3. Notation like the gradient $\nabla$ and its adjoint $div$ could be briefly introduced for the graph domain.

## References:

[1]: Lin, Xixun, et al. "Graph neural stochastic diffusion for estimating uncertainty in node classification." Forty-first International Conference on Machine Learning. 2024.
[2]: Fuchsgruber, Dominik, Tom Wollschläger, and Stephan Günnemann. "Energy-based Epistemic Uncertainty for Graph Neural Networks." The Thirty-eighth Annual Conference on Neural Information Processing Systems.

---

> ### Author Rebuttal · Authors · 2025-07-30
>
> We thank the reviewer for the comprehensive set of feedbacks and questions, and here we attempt to address your questions and concerns. We refer to reviewers' raised weaknesses by __W1-W8__ and questions as __Q1-Q4__.
>
> __W1 (novelty)__: While building on GNSD's foundation, our contributions are substantial. GNSD uses a fixed graph Laplacian with limited insights on noise propagation on graph. By embedding a noise process with covariance structure through the $\Phi$ transformation, our approach differs from GNSD with inductive bias and explicitly models the correlation structure of uncertainty. It is a fundamental change in the noise process that enables us to model heterophilic (low LI) uncertainty patterns that GNSD cannot capture, and its implementation in section 3.3 is derived from the theory of semilinear SPDEs in stochastic analysis, which allows a random graph ODE approximation using the semigroup and Wong-Zakai theorem, rather than a simple latent graph SDE in GNSD, which lacks a clear connection with the SPDE representation itself.
>
> __W2 (background)__: we acknowledge that the paper does require some additional backgrounds for Stochastic Partial Differential Equations, and due to the page limit, we put extensive background materials in Appendix A. In the camera-ready version, we will modify line 51-59 of the manuscript to add a description of the method developed in the paper. Here is the content:
>
> _The flow of the paper goes like the following: we first motivate a noise process with a spatial kernel structure that can capture the smoothness of correlation between uncertainty of graph nodes, then we principally derive an end-to-end message passing scheme on graph that utilizes this noise process using the theory of SPDE._
>
> The reviewer's understanding of the flow is mainly correct, with the caveats that we refrain from talking about aleatoric and epistemic uncertainty explicitly, which we find to be defined differently across existing literature. The intuition of frequency in terms of Laplacian eigenvalues is also relevant, in the sense that the Matérn kernel magnifies graph signal's smoothness for large $\nu$ and produces rough dependencies with smaller $\nu$.  The spectral approach for SPDE and Wiener process connects more directly with the Hilbert space theory (the Karhunen-Loeve expansion in definition 2) for representing stochastic processes rather than frequency domain analysis of signals, and we motivate it in the paper from the former perspective. However, the reviewers comment from the signal processing perspective is a promising direction to look at the current framework.
>
> __W3,4 ($\Phi$ and Matérn kernel)__: Thanks for the question. In section 3.1 we are motivating a general theory of $\Phi$-Wiener process, which uses a general transformation $\Phi$, and the Matérn kernel is a special case of the transformation. We touched upon this connection in section 2.4, and in the rest of the paper, we confine ourselves to use only the Matérn kernel. The reason why we want to develop a general framework for $\Phi$ is to motivate further studies using different kernel functions that can also lead to meaningful solution of SPDEs, which can be left for further work. Nevertheless, the condition of $\Phi(\Delta)$ being trace-class allows us to develop the current framework, and the Matérn kernel satisfies the condition and is known solution of SPDE, with relevant backgrounds mentioned in section 2.4 of the paper.
>
> __W5 (Figure 1)__:  Figure 1 is intended to show the need for a flexible spectral transformation. The same heterophilic graph requires fundamentally different uncertainty correlation patterns depending on the underlying label structure. Traditional methods are locked into topology-based correlations, while our $\Phi$-wiener framework with tunable $\nu$ allows adaptation to the appropriate correlation structure. The red edges in the small-$\nu$ limit show correlations that would be impossible with the fixed Laplacian but may be exactly what's needed for heterophilic uncertainty patterns. This is also connected to the observation made in section 4.3, which views the covariance structure as dynamic graph rewiring.
>
> __W6 (Grammar)__: We thank the reviewer for pointing these out. We will review the paper thoroughly for grammatical errors and other notational inconsistencies.
>
> __W7 (GEBM baseline)__: Thanks for the valuable suggestion. With some major efforts, we have now successfully added GEBM to the list of baseline models. We use the hyper-parameter set up in Appendix B.2 of the GEBM paper with the multiscale energy functions described in the paper. We ran the experiments on each dataset 5 times with different seed values, under the 3 types of OOD shifts. Here are the results of this model on the proposed datasets, which differ from the datasets settings used in GEBM. We will include this model's results in the camera-ready version of our paper, in section 4. To summarize the results, our model outperforms GEBM on all datasets except Citeseer, and we found that GEBM gives pathological behavior on the questions dataset for the structural perturbation case, where it produces higher energy score for IND samples than OOD samples.
>
> | Dataset | Label OOD | | | Feature OOD | | | Structure OOD | | |
> |---------|-------------------|---|---|---------------|---|---|-----------|---|---|
> |         | **AUROC** | **Acc** | **FPR95** | **AUROC** | **Acc** | **FPR95** | **AUROC** | **Acc** | **FPR95** |
> | **cora** | 90.50±11.23 | 79.90±10.15 | 30.80±13.73 | 75.00±7.81 | 73.20±2.73 | 61.00±2.79 | 78.50±9.95 | 77.70±5.51 | 51.00±8.29 |
> | **pubmed** | 79.50±2.36 | 64.60±0.79 | 64.30±7.16 | 94.20±2.48 | 90.30±2.48 | 14.90±9.06 | 94.70±1.70 | 91.40±1.21 | 12.70±3.28 |
> | **citeseer** | 94.20±4.52 | 88.60±5.32 | 26.90±13.26 | 82.80±2.55 | 75.40±1.39 | 57.30±4.21 | 80.80±8.00 | 75.90±5.66 | 77.40±3.08 |
> | **roman_empire** | 57.80±2.90 | 54.60±2.33 | 89.60±0.35 | 60.80±1.08 | 57.90±0.82 | 87.90±0.98 | 49.60±0.38 | 53.70±0.21 | 99.90±0.04 |
> | **tolokers** | 56.80±0.72 | 53.00±0.32 | 96.60±0.26 | 5.00±1.44 | 50.50±0.16 | 97.50±0.79 | 1.80±0.30 | 50.90±0.17 | 98.20±0.30 |
> | **minesweeper** | 59.00±3.76 | 54.60±2.18 | 92.20±0.64 | 51.60±13.96 | 58.80±3.44 | 80.10±7.83 | 42.60±16.22 | 64.90±6.27 | 66.20±13.76 |
> | **questions** | 51.60±0.15 | 50.40±0.00 | 97.90±0.13 | 67.20±0.72 | 66.70±0.71 | 91.20±0.30 | 0.00±0.00 | 50.00±0.00 | 100.00±0.00 |
> | **amazon_ratings** | 57.20±2.30 | 52.60±1.01 | 85.90±1.66 | 51.20±4.00 | 61.10±2.31 | 74.90±4.37 | 35.90±6.29 | 61.10±3.21 | 73.50±6.64 |
>
> __W8 (introducing $\nabla$ on graph)__:  Due to the limited space, we formalized the notations in Appendix A.1. We will add these definitions more systematically in the Appendix and reference them appropriately to make the paper more self contained.
>
> __Q1 ($\nu$ tuning)__: The $\nu$ parameter is tuned as indicated in Appendix C.6.
>
> __Q2 ($\nu$ interpretation)__: The $\Phi$ function provides the general framework for the spectral transformation of the noise process while $\nu$ controls the smoothness of the specific functional form of our noise process, the Matérn kernel. In a sense, they serve complementary roles because $\Phi$ enables the mathematical framework, while $\nu$ specifies the smoothness needed for LI-related adaptations. _Long-range_ refers to noise correlations between topologically distant nodes on the graph, which is appropriate since low $\nu$ enables correlations between nodes that are far apart in graph structure. The intuition of frequency in terms of Laplacian eigenvalues is also relevant, as the Matérn kernel magnifies graph signal's smoothness for large $\nu$ and produces rough dependencies with smaller $\nu$.  The spectral approach for SPDE and Wiener process connects more directly with the Hilbert space theory (Karhunen-Loeve expansion in definition 2) for representing stochastic processes, rather than frequency domain analysis of signals, and we motivate it in the paper from the former perspective. However, the reviewer's comment is a promising direction to look at the current framework.
>
> __Q3 (two class scenario)__: Thanks for the insightful question. This question illustrates the difference between the task of node classification and OOD detection. On the high level, the question/concern is valid if we are working with a node classification task, which uses the largest logit value to indicate label prediction. Instead, we use the overall entropy of the class distribution to distinguish between IND and OOD samples at test time. If a deterministic node classification model only sees one class at training, it will simply overfit to that one class using cross entropy loss, but our probabilistic model can handle the existence of the other class at test time, precisely because of the uncertainty it induces, since the OOD samples' predicted class probabilities can have a lot higher entropy. In short, for OOD detection, the model sees the _normal_ and learns what the normal looks like at training time and detect the _abnormal_ at test time from normal and abnormal samples. Some previous works have explored the case of OOD samples exposure during training time, which allows them to further the gap between IND and OOD sample's entropy or energy function, but in our case we choose to handle the more general scenario of no OOD exposures during training. This point was touched upon briefly in Appendix C.4, but we will move it to the experimental section and make it more clear.
>
> __Q4 (LI calculation)__: LI is computed using empirical label distributions estimated from the training portion of each dataset. This follows the methodology of [5]. We will rework section 2.2 to make this point more clear.
>
> We again thank the reviewer for their comprehensive feedbacks and questions. We believe that by addressing the above weaknesses, the work will be presented more clearly. Please let us know if we have not fully addressed your concerns.

---

> ### Comment · Reviewer_iXcv · 2025-08-04
>
> Thank you for your thoughtful response!
>
> W1: I understand that generalizing to arbitrary transformations $\Phi$ is sensible, but I still view it as somewhat incremental. However, I appreciate the theoretical discussion and motivation of this concept: The paper makes a valuable contribution to the field.
>
> W2: I appreciate the additional clarifications, even though I do not believe they significantly solve the problem. At the same time, I also understand that the theoretical background the paper requires can not be presented thoroughly given the page limit. The provided Appendix seems to be an appropriate starting point.
>
> W3 & 4: This connection was not clear to me from the manuscript. I strongly encourage the authors to clarify that while the theory is concerned with arbitrary non-linear transformations, in practice, the evaluation resorts to the well-known Matern kernel. While I see the value of the more general theoretical derivation, I would also appreciate the transparency regarding the implementation -- especially as the theoretical discussion suggests more general $\Phi$. This also can help to guide the reader through the paper -- indirectly addressing W2.
>
> W5: Thank you for the clarification. I think the Figure is a nice visualization if this information is explained similarly in the text. Especially Fig. 1a, however, is not discussed sufficiently there. The paper could benefit from a proper explanation of Figure 1 in the main text -- ideally already in the introduction where the figure is located.
>
> W7: Thank you for taking this effort. The results look very convincing to me!
>
> W8: Thank you for adding this clarification to the main text in your revision!
>
> Furthermore, Q1-Q4 are clear to me now -- thank you for the thorough elaboration.
>
> Overall, I recognize the author's effort to address my concerns. Especially my minor concerns and questions have been well resolved. However, my reservations both regarding the novelty and clarity of the paper partially remain. Since I still view this paper as a valuable contribution to the field, I will increase my score accordingly.

---

> > ### Author Response · Authors · 2025-08-04
> >
> > Thanks a lot for your acknowledgement and support! We will surely address the clarity issues in our camera-ready version of the paper!

---

### Official Review · Reviewer_TK1Y · 2025-06-07

**Clarity:** 3
**Significance:** 3
**Originality:** 3
**Rating:** 5
**Confidence:** 4

**Summary:**

The authors propose to model message passing in a graph as a stochastic partial differential equation (SPDE) with the flexible covariance structure of the Brownian noise (here realized by a Matérn kernel). This allows for better control over defining the spatial correlation pattern that suits the graph structure, improving the uncertainty estimation and OOD detection performance.

**Questions:**

- High correlation between nodes (low $\nu$) comes with high node variance. In such case, what is the strategy to balance those? And how careful should we be to choose the OOD uncertainty threshold in cases with low $\nu$ ?

**Ethical Concerns:**

["NO or VERY MINOR ethics concerns only"]

**Final Justification:**

This is a good paper. I had minor clarification concern regarding how the variance of nodes would be effected by $ \nu $, which the reviewers have duly addressed.

**Limitations:**

Yes

**Quality:**

4

**Strengths And Weaknesses:**

**Strength:**
- The idea is well-motivated and theoretically sound.
- Proposed method outperforms state-of-the-art uncertainty estimation baselines in graphs.

**Weakness:**
- No significant weakness
- To better gain insights in the results presented in Figure 2(b), it would be useful to see the some kind of LI distribution of nodes in the graphs having high LI (e.g Citeseer), medium LI (Pubmed) and low LI(Questions).

---

> ### Author Rebuttal · Authors · 2025-07-30
>
> We thank the reviewer for the positive remarks and for the questions raised. Here we attempt to address the concerns and questions:
>
> __To better gain insights in the results presented in Figure 2(b), it would be useful to see the some kind of LI distribution of nodes in the graphs having high LI (e.g Citeseer), medium LI (Pubmed) and low LI(Questions).__
>
> In our case, Label Informativeness (LI) is a graph-level metrics since its computation depends on random sampling of the edges of graph, hence it is not directly defined for each node, unlike the traditional but limited homophily and heterophily metrics. To fulfill the reviewer's request, we can randomly sample edges, compute edge LI, and marginalize to obtain node LI, but this would not be the same metrics used in the paper.
>
> __High correlation between nodes (low $\nu$) comes with high node variance. In such case, what is the strategy to balance those? And how careful should we be to choose the OOD uncertainty threshold in cases with low $\nu$ ?__
>
>  Low $\nu$ in our case corresponds to low edge LI, and the covariance structure of the Mat\'ern kernel is able to handle the situation automatically due to the roughness of the kernel. As for the threshold for calculating detection accuracy, we first take the range of scores of IND and OOD samples generated by the models, then take the detection accuracy as the average across 10 equally sized threshold values in between. For each threshold we calculate the accuracy and take the \textit{minimum} accuracy to report. Otherwise, we use AUC to make the reporting independent of threshold choice. In other words, the evaluation method is not $\nu$ dependent and is consistent for all the models in comparison.
>
>
> We again thank the reviewer for the helpful comments and questions.  We believe that by addressing the above weaknesses, the work will be presented more clearly. Please let us know if we have not fully addressed your concerns.

---

> > ### Comment · Reviewer_TK1Y · 2025-08-04
> > **Calrification**
> >
> > **Low $\nu$ in our case corresponds to low edge LI, and the covariance structure of the Mat'ern kernel is able to handle the situation automatically due to the roughness of the kernel.**
> >
> > I am having difficulty in understanding this point. Could you explain more? My question was that when there is high correlation between nodes, the node variances are high, meaning that the uncertainty over the nodes in the graph is high, which is probably not desirable. How does the Mat'ern Kernel control that?

---

> ### Author Response · Authors · 2025-08-04
>
> Dear reviewer, thanks for the follow-up question! I see that you are asking about the variance of the noise process on each node itself, rather than the covariance structure. Let me answer it in this way:
>
> For the function $f(\lambda) = \left(\frac{2\nu}{\kappa^2} + \lambda \right)^{-\nu}$ and the graph Matern kernel: $K = f(L) = U f(\Lambda) U^\top = U \left(\frac{2\nu}{\kappa^2}I + \Lambda\right)^{-\nu} U^\top$, the variance of each code can be written as
> $$
> K_{ii}= (Uf(\Lambda) U^T)\_{ii} = \sum_{j=1}^{n} f(\lambda_j) U_{ij}^2 = \sum\_{j=1}^n (\frac{2\nu}{\kappa^2} + \lambda\_j)^{-\nu} U\_{ij}^2
> $$
>
> This variance depends on the graph topology as the above equation is a function of Laplacian eigenvalues and eigen-vectors, and is not directly clear whether they are small or large by just looking at $\nu$. However, what we can tell is that, fixing all the other terms constant and set $2\nu/\kappa^2 = 1$, then since graph Laplacian eigenvalues are non-negative, the larger the value of $\nu$, the more it seems to penalize the variance term (hence the terminology of a smooth kernel). In addition, if $\nu$ is large, the distribution of variance is more even for neighbors or connected components (if one node has large variance, neighbors also do), whereas if $\nu$ is small, we can have two distant nodes sensitive to the variance change of each other.  Ultimately the kernel function controls covariance while variance has more to do with graph topology itself, due to the direct link with $\Lambda$, $U$.
>
> Hence, in our model, the more direct answer to how variance is controlled for each node _individually_ is the integration time in the graph Neural ODE in section 3.3, since longer integration time leads to accumulation of more variance, hence more uncertainty.
>
> The point raised by you is a very interesting direction to explore, since we know that more layers of GNN (corresponding to more integration steps) usually lead to over-smoothing of graph signals, but the noise accumulation can seem to introduce variance. Since these two dynamics can interact with each other in intricate ways, it may even be possible that longer integration time doesn't necessarily lead to exploding variance. In our empirical studies, we concluded that learning converges with loss function converging, but it would indeed be interesting to study how the variance of each individual nodes (or sub-regions, since the graphs are usually very large) evolve. We believe that this topic itself can be an interest of extensive future studies.

---

> > ### Comment · Reviewer_TK1Y · 2025-08-06
> >
> > Thank you. I think this detail would help readers if included in the paper or the appendix.
> >
> > I stand with my rating.

---

> > > ### Author Response · Authors · 2025-08-06
> > >
> > > Dear reviewer, thank you so much for the acknowledgment and the support! We deeply appreciate the time and effort you have invested in reviewing our work and offering these detailed comments. Your feedback has been instrumental in enhancing the quality of our work. We will enrich the appendix section per our discussions.

---

### Official Review · Reviewer_9pWG · 2025-07-02

**Clarity:** 2
**Significance:** 3
**Originality:** 2
**Rating:** 4
**Confidence:** 3

**Summary:**

This paper proposes a novel Structure Informed Stochastic Partial Differential Equation (SISPDE) framework for uncertainty estimation on graphs. The key innovation lies in modeling spatially correlated noise via a Matérn-Gaussian-Process-driven SPDE, which generalizes prior Q-Wiener processes by explicitly controlling covariance smoothness through parameter ν. This enables superior uncertainty quantification under both high and low label informativeness (LI) scenarios. Theoretical contributions include proofs for the Φ-Wiener process properties (Theorems 1–2), existence/uniqueness of SPDE solutions (Theorem 3), and Chebyshev approximation bounds (Theorem 4). Experiments on 8 graph datasets demonstrate state-of-the-art OOD detection, especially on low-LI graphs where traditional GNNs struggle.

**Questions:**

1)​​Kernel Generalization:​​ Why not explore non-Matérn kernels (e.g., RBF)? If ν is pivotal for LI adaptation (Fig 2b), does this limit applicability to non-Matérn covariances?

​​2)Smoothness vs. LI:​​ Can you establish a quantitative relationship between optimal ν and dataset LI (e.g., via regression)? Fig 2b implies heuristic tuning.

​​3)Scalability:​​ For large graphs (e.g., >10^6 nodes), does Chebyshev approximation (Theorem 4) suffice? Have you tested scalability vs. graph size?

**Ethical Concerns:**

["NO or VERY MINOR ethics concerns only"]

**Final Justification:**

I stand with my rating.

**Limitations:**

yes

**Quality:**

2

**Strengths And Weaknesses:**

Quality

​​Strength:​​ Rigorous theoretical grounding with 4 theorems and appendices. Extensive experiments (8 datasets, 3 OOD types, 5 seeds) validate claims.

​​Strength:​​ Matérn kernel’s ν adapts to LI – low ν captures long-range dependencies in heterophilic graphs (Figure 2b), addressing a critical gap.
​​

Weakness:​​ Computational cost of Cholesky decomposition for Matérn kernel not deeply analyzed (runtime in Fig 4 but scalability limits unclear).

​​Clarity


​​Strength:​​ Well-structured with clear analogies between SPDEs and GNN message passing (Section 2.3–2.4).
​

​Weakness:​​ Appendix-heavy proofs may disrupt flow; some notation (e.g., Φ-Wiener process in Def 2) requires multiple reads.


Significance


​​Strength:​​ Unifies stochastic processes (GNSD), graph kernels (Matérn GP), and PDE-based GNNs (GRAND) into a flexible framework.
​​

Strength:​​ Achieves SOTA on low-LI graphs (Table 1), where uncertainty estimation is most challenging (Section 4.2).



Originality


​​Strength:​​ First to integrate SPDE-driven spatially correlated noise with tunable smoothness for graph uncertainty. The physics-inspired approach (Figure 1b) is novel.

---

> ### Author Rebuttal · Authors · 2025-07-30
>
> We thank the reviewer for the valuable feedbacks and questions. Here we attempt to address your questions and concerns below:
>
> __W1: Appendix-heavy proofs may disrupt flow.__
>
>  Due to page limit, we chose to put the proofs in the Appendix section. In the camera-ready version, we will add sketch proof in the paper. For example, for Theorem 1, we will add the following below the theorem, at line 170:
>
> __Sketch Proof__: the result follows from the orthogonality of the eigenvectors of the graph Laplacian and the fact that Wiener processes are zero-centered with correlation structure of $ Cov(W(t),W(s)) = min(t, s)$  and $W(t)-W(s) \sim \mathcal{N}(0,t-s)$ for $t>s$.
>
> __W2: Some notation (e.g., $\Phi$-Wiener process in Def 2) requires multiple reads.__
>
> In the camera-ready version, we will add a short clarification part for the relationship between $\phi$ and $\Phi$, a function and its matrix functional form, which we believe could be where the reviewer find to be unclear. In particular, we will add the notation of $\Phi(\Delta) =  \text{diag}([\phi(\lambda_1), \cdots,\phi(\lambda_n)])$ for diagonal $\Delta$ and $\Phi(X) = U \Phi(\Lambda)U^T$ for $X=U\Lambda U^T$ in section 2.4, at line 125 of the manuscript.
>
> __Q1: kernel Generalization: Why not explore non-Matérn kernels (e.g., RBF)? If $\nu$ is pivotal for LI adaptation (Fig 2b), does this limit applicability to non-Matérn covariances?__
>
> The RBF kernel can be seen as a special case of theMatérn kernel, that is, when $\nu \rightarrow \infty$. So the RBF kernel does serve as an extreme case for studying the smoothness parameter $\nu$, for which we provide the additional results in the following table that explicitly uses the RBF kernel. We report the mean and standard deviation over 5 runs for the same set of experiments as other baseline models in the paper, and will add it in the camera-ready version as part of the discussion of the parameter $\nu$. In the camera ready version, in section 4.2 below line 320, we will add further observation of the result from a smooth kernel (RBF) and state the superiority of the more general Matérn kernel, which gives us control of the smoothness explicitly.
>
> | Dataset | **LABEL OOD** | | | **FEATURE OOD** | | | **STRUCTURE OOD** | | |
> |---------|---------|---------|---------|---------|---------|---------|---------|---------|---------|
> | | __AUC__ | __DET-ACC__ | __FPR95__ | __AUC__ | __DET-ACC__ | __FPR95__ | __AUC__ | __DET-ACC__ | __FPR95__ |
> | **cora** | 95.220 ± 0.536 | 88.899 ± 0.752 | 23.671 ± 5.615 | 91.375 ± 0.835 | 85.302 ± 1.150 | 61.573 ± 8.031 | 83.546 ± 2.241 | 77.928 ± 2.609 | 77.629 ± 6.857 |
> | **pubmed** | 79.157 ± 5.815 | 73.594 ± 4.623 | 62.915 ± 12.068 | 90.972 ± 5.786 | 85.612 ± 5.642 | 51.302 ± 32.345 | 95.185 ± 1.907 | 89.559 ± 1.757 | 23.168 ± 20.215 |
> | **citeseer** | 80.259 ± 2.308 | 73.828 ± 2.520 | 62.951 ± 5.878 | 83.453 ± 3.574 | 77.554 ± 3.257 | 80.331 ± 12.477 | 73.641 ± 2.736 | 69.775 ± 1.911 | 88.500 ± 8.154 |
> | **roman_empire** | 51.757 ± 0.565 | 51.630 ± 0.290 | 93.240 ± 0.285 | 51.500 ± 0.259 | 51.340 ± 0.151 | 94.129 ± 0.080 | 56.582 ± 0.677 | 57.800 ± 0.551 | 81.935 ± 0.921 |
> | **minesweeper** | 53.998 ± 3.206 | 53.695 ± 2.160 | 92.580 ± 0.952 | 55.168 ± 3.243 | 56.600 ± 2.221 | 96.014 ± 2.212 | 52.967 ± 1.654 | 56.845 ± 1.412 | 95.924 ± 1.189 |
> | **tolokers** | 60.632 ± 8.902 | 59.263 ± 6.164 | 92.362 ± 7.944 | 76.620 ± 13.619 | 74.516 ± 12.588 | 76.503 ± 33.647 | 93.115 ± 1.226 | 93.986 ± 3.898 | 99.980 ± 0.041 |
> | **questions** | 54.297 ± 0.968 | 54.446 ± 1.383 | 94.466 ± 0.430 | 50.377 ± 0.040 | 50.483 ± 0.059 | 94.614 ± 0.135 | 53.908 ± 0.718 | 61.400 ± 0.391 | 99.534 ± 0.022 |
> | **amazon_ratings** | 52.616 ± 0.489 | 52.310 ± 0.464 | 93.513 ± 0.508 | 51.658 ± 0.320 | 51.367 ± 0.307 | 94.010 ± 0.409 | 88.123 ± 0.071 | 87.262 ± 0.359 | 89.534 ± 0.342 |
>
> To summarize the results above, the RBF kernel performs decently (ranked 3rd generally) for the high LI datasets, but performed poorly on the low LI datasets, which agrees with the current conclusions we draw from $\nu$ value and LI.
>
> Moreover, our construction in section 3.1 and 3.2 indicates that as long as the kernel function for the Gaussian process is trace class, the SPDE should have weak solution, which means we can indeed extend it to Kernels that are non-Matérn. Parameter $\nu$ is an artifact of the Matérn kernel, so other kernel functions may have their own specific parameters.
>
>
> __Q2. Smoothness vs. LI: Can you establish a quantitative relationship between optimal $\nu$ and dataset LI (e.g., via regression)? Fig 2b implies heuristic tuning.__
>
> Due to the constraint of NeurIPS rebuttal, we cannot provide a regression figure. However, we choose to provide the data points used for a scatter plot, which should clearly indicates a positive correlation between LI and optimal $\nu$ values overall. Here are the two lists: LI = [0.590, 0.409, 0.451, 0.110, 0.007, 0.000, 0.001, 0.040] and $\nu$ = [14.67, 9.33, 18.33, 1.30, 0.57, 0.97, 0.23, 0.90]. They have a Pearson correlation of 0.939 with a p-value of 0.00055, indicating strong linear correlation.
>
> __Scalability: For large graphs (e.g., $>10^6$ nodes), does Chebyshev approximation (Theorem 4) suffice? Have you tested scalability vs. graph size?__
>
> The Chebyshev approximation has the runtime complexity of $\mathcal{O}(m|E|)$ where $m$ is the degree of the polynomial and $|E|$ is the number of edges. In our camera ready version, we will add this previously known result at line 237. In our Appendix section C.7 and figure 4, we have provided a runtime analysis for the purpose of studying scalability w.r.t the other baselines, and we found that indeed the performance of the Chebyshev approximation lags when the graph is dense. Since in general  $O(K|E|) < O(|V|^3)$, the performance is bounded by the number of edges and much less expensive than directly computing the eigendecomposition of the graph Laplacian.
>
>
> We again thank the reviewer for their insightful feedbacks and questions. We believe that by addressing the above weaknesses, the work will be presented more clearly. Please let us know if we have not fully addressed your concerns.

---

> ### Comment · Area_Chair_616S · 2025-08-06
>
> Please respond to the authors' rebuttal and indicate if you are happy to reconsider your score based on the rebuttal and the other reviewers' comments.

---

### Official Review · Reviewer_Ewp5 · 2025-07-03

**Clarity:** 3
**Significance:** 3
**Originality:** 3
**Rating:** 4
**Confidence:** 4

**Summary:**

This paper proposes a new method of uncertainty estimation on graphs by making an analogy between the evolution of SPDE and message passing. Through introducing the noise process, the proposed method enables GNNs to quantify uncertainty and to achieve detection tasks on graphs. The core innovation is proposing a new noise form based on graph Gaussian Process to learn the spatial-temporal covariance of noise. Experimental results demonstrate the effectiveness of the proposed methods.

**Questions:**

1. The paper mentions uncertainty estimation but does not explain how the two types of uncertainty are accurately learned after formulating the modeling equation. Therefore, how are epistemic uncertainty and aleatoric uncertainty actually learned?

2. Besides OOD detection, does uncertainty estimation serve any other purposes?

**Ethical Concerns:**

["NO or VERY MINOR ethics concerns only"]

**Final Justification:**

Most of the concerns have been addressed, but I still have some reservations regarding the novelty. As a result, I have increased the score by just one point, from 3 to 4.

**Quality:**

3

**Strengths And Weaknesses:**

Strengths
1. Exploring the integration of SPDEs with GNNs is a highly interesting and promising direction. The authors propose a novel form of SPDE and provide a theoretical analysis of it.
2. The authors provide a detailed descriptions of graph SPDE and related works, which clarifies the logical flow of the paper and enhances its readability.

Weaknesses
1. The novelty of the proposed model appears to be limited. The combination of SPDEs and GNNs has already been explored in prior works [1,2], and this paper mainly makes adjustments to the noise term. Moreover, the dependence of the noise can also be learned through the stochastic forcing term, rather than simply adding independent noise. Therefore, the authors need to place greater emphasis on this aspect.
2. The computation of graph Gaussian processes is typically very complex. Therefore, the authors should clarify the computational complexity of the proposed model and compare its efficiency with that of other methods.
3. The connection between label informativeness and the proposed method in this paper is not clearly emphasized. It is necessary to further highlight the relationship between label informativeness and the proposed approach.

[1] Graph Neural Stochastic Diffusion for Estimating Uncertainty in Node Classification. Xixun Lin et al. ICML 2024.

[2] Uncertainty Modeling in Graph Neural Networks via Stochastic Differential Equations. Richard Bergna et al. ICLR 2025.

---

> ### Author Rebuttal · Authors · 2025-07-30
>
> We thank the reviewer for the suggestions and the here we attempt to address your concerns:
>
> __W1. The novelty of the proposed model appears to be limited. The combination of SPDEs and GNNs has already been explored in prior works [1,2], and this paper mainly makes adjustments to the noise term. Moreover, the dependence of the noise can also be learned through the stochastic forcing term, rather than simply adding independent noise. Therefore, the authors need to place greater emphasis on this aspect.__
>
> We respectfully disagree that our novelty is limited. First, to clarify, despite some limited degrees of similarity in the  model implementation, our work is fundamentally different from [2], where a latent graph SODE is used with the learning framework formulated using variational inference, similar to the same authors' work in [4], which the latent SODE was used for time series data. Our model focuses on studying message passing on graph data with explicit spatially-correlated noise structures, hence SPDE-motivated, and we are not concerned with sequential or time series modeling, although we think this task is an interesting one to extend our model to. Also from section 3.3, we implement the model as a randomly forced ODE, which is principally motivated by semigroup theory and  Wong-Zakai approximation of SPDE used in [3], rather than proposing a latent SDE model, as was done in [2] and [4]. Therefore, we have provided a much more natural and principled approach compared to [2]. We will add the clarification with the added reference to [2] in our section 2.1 in the camera-ready version. We thank the reviewer for raising the concern.
>
> [3]. Cristopher Salvi, Maud Lemercier, Andris Gerasimovics: Neural Stochastic PDEs: Resolution-Invariant Learning of Continuous Spatiotemporal Dynamics. NeurIPS 2022.
>
> [4]. Richard Bergna, Felix L. Opolka, Pietro Liò, José Miguel Hernández-Lobato: Graph Neural Stochastic Differential Equations. CoRR abs/2308.12316 (2023).
>
> Second, for the GNSD model: While prior work such as GNSD [1] did explore the SPDE-GNN connection, our contributions are fundamentally different in three ways.
>
> 1. Spatial correlation modeling, which we believe is not just a simple adjustment to the noise term. We model spatially correlated noise through the use of a Matérn gaussian process, which explicitly gives us control of strength of correlation between uncertainty at each node. In [1], the Wiener process was not directly transformed but the heavy lifting was done using an additional diffusion network $G$ without any inductive bias. We don't want to claim that learning $G$ would be impossible, but we suspect that the learning task would be significantly more difficult (our empirical results support that, since we tuned the hyperparameters of both models).
>
> 2. Our framework's derivation as randomly forced graph ODE, rather than a latent SDE like in [1], is more principled, using the semigroup and Wong-Zakai approximation of SPDE, instead of simply proposing a neural SDE model as the actual implementation, as was done in [1]. In the camera-ready version, we will add the clarification as part of section 2.1, talking about the latent graph SDE model in the context of sequence data modeling and how our model differs from it.
>
> 3. We provide both a theoretical motivation and empirical validation showing the connection between kernel smoothness, $\nu$, and the label informativeness. Such a connection moves beyond the traditional homophilic/heterophilic paradigm and was not addressed before in [1]. Specifically, we find significant variations in the required $\nu$ parameter among the heterophilic/low LI  datasets (roman-empire, minesweeper, amazon ratings, tolokers, questions) and a significant gap between the optimal $\nu$ values for low LI and high LI datasets. Further, we observe that low $\nu$ is required to capture long-range uncertainty that traditional methods cannot capture. We believe that drawing and strengthening this connection is a novel synthesis and contributes meaningfully to the work.
>
> 4. We introduce the $\Phi$-Wiener process that generalizes the $Q$-Wiener process. This process generalizes the $Q$-Wiener process through a spectral transformation, and goes beyond merely ``adjusting the noise term'' but instead allows us to develop a principled mathematical framework. We believe that the natural extension as seem by the reviewer is an artifact of the elegance of mathematics, rather than a limit on novelty.
>
> Compared to our work, the relationship between [1] and [2] is closer in the sense that the implementation of [1] closely follows [2], a latent SDE, despite the motivation of using SPDE in [1]. Our work correctly motivates the neural implementation based on the work from [3] using sound mathematical principles, which [1] failed to do, and our implementation differs from [1] because of that.
>
> __W2. The computation of graph Gaussian processes is typically very complex. Therefore, the authors should clarify the computational complexity of the proposed model and compare its efficiency with that of other methods.__
>
> We address the computational complexity through the Chebyshev approximation (Theorem 4). This reduces the $O(|V|^3)$ eigen decomposition to $O(m |E|)$ operations, where $m$ is the polynomial degree. The error bound shows exponential convergence for smooth kernels. Further, we present a comparison of runtimes in Figure 4 in the appendix. In Figure 4 we see that aside from pubmed and questions, our model is similar in computational cost to the other baselines.
>
> __W3. The connection between label informativeness and the proposed method in this paper is not clearly emphasized. It is necessary to further highlight the relationship between label informativeness and the proposed approach.__
>
> We agree that the connection between label informativeness is both motivating (line 86-91) and empirical, but lacks theoretical rigor. We do empirically confirm these results in Figure 2(b). Further, we validate this empirical evidence in Table 1 by showing that our method out-performs the benchmarks. We believe that this evidence is enough to justify the connection between LI and SISPDE. In terms of writing, we will surely highlight the relationship further in our camera-ready version. We thank the reviewer for the constructive feedback.
>
> __Q1. The paper mentions uncertainty estimation but does not explain how the two types of uncertainty are accurately learned after formulating the modeling equation. Therefore, how are epistemic uncertainty and aleatoric uncertainty actually learned?__
>
> Our method estimates both types of uncertainty, but we opted to not decompose the uncertainty into epistemic and aleatoric uncertainty since we noticed differing definitions in the literature and a lack of consensus. That being said, one sensible way to understand this is that Aleatoric uncertainty is captured through the stochastic noise term which represents the inherent randomness in the data. The epistemic uncertainty could be quantified through the distribution of final states and measured through the entropy. Here too, we note that spatially correlated noise better captures the true Aleatoric uncertainty, especially for low-LI graphs where neighboring nodes may have different uncertainty patterns.
>
> __Q2. Besides OOD detection, does uncertainty estimation serve any other purposes?__
>
> Our uncertainty framework naturally extends to other applications, including:
> - Active learning for node labeling. Here we would pick the most uncertain vertices to send out for additional labeling
> - Confidence aware predictions
> - Robust predictions in the presence of distributional shifts, either in the graph structure or in the feature distributions.
> We mentioned some of these directions in future work, such as extending the approach to model stochastic spatial dynamical systems, and are excited about their prospect. We ultimately decided to not include those applications in this work because we believed that this work was self contained with the theoretical developments and the OOD application. In the camera-ready version, we will add the above applications in the last part of section 5.
>
> We again thank the reviewer for their insightful feedbacks and questions. We believe that by addressing the above weaknesses, the work will be presented more clearly. Please let us know if we have not fully addressed your concerns.

---

> > ### Comment · Reviewer_Ewp5 · 2025-08-06
> >
> > Thank you for your reply. The reply has addressed my concerns. I will raise the rating from 3 to 4.

---

### Decision · Program_Chairs · 2025-09-17

**Decision:**

Accept (poster)

**Comment:**

Summary of the paper:

The paper proposes a novel Structure Informed Stochastic Partial Differential Equation (SISPDE) framework for uncertainty estimation on graphs. The key innovation lies in modeling spatially correlated noise via a Matérn-Gaussian-Process-driven SPDE, which generalizes prior Q-Wiener processes by explicitly controlling covariance smoothness through parameter. This enables superior uncertainty quantification under both high and low label informativeness (LI) scenarios. Theoretical contributions include several proofs for properties, existence/uniqueness of SPDE solutions, and Chebyshev approximation bounds. Experiments on 8 graph datasets demonstrate state-of-the-art OOD detection, especially on low-LI graphs where traditional GNNs struggle.

Summary of the discussion:

All the reviewers agree on acceptance. Two reviewers still have concerns regarding the novelty. One still has concerns regarding the clarity of the paper. One of the reviewers had minor clarification concern regarding how the variance of nodes would be effected by the model's parameter. This was successfully addressed by the authors.

Recommendation:

All reviewers vote for acceptance. I, therefore, recommend acceptance and encourage the authors to use the feedback provided to improve the paper for its final version, especially, to address the remaining concerns of the reviewers.